# Graphon Neural Differential Equations and Transferabilty of Graph Neural Differential Equations

## Abstract

Graph Neural Differential Equations (GNDEs) extend Graph Neural Networks (GNNs) to a continuous-depth framework, providing a robust tool for modeling complex network dynamics. In this paper, we investigate the potential of GNDEs for transferring knowledge across different graphs with shared convolutional structures. To bridge the gap between discrete and continuous graph representations, we introduce Graphon Neural Differential Equations (Graphon-NDEs) as the continuous limit of GNDEs. Using tools from dynamical system theories and graph limit theory, we rigorously establish this continuum limit and develop a mathematical framework to quantify the approximation error between a GNDE and its corresponding Graphon-NDE, which decreases as the number of nodes increases, ensuring reliable transferability. We further derive specific rates for various graph families, providing practical insights into the performance of GNDEs. These findings extend recent results on GNNs to the continuous-depth setting and reveal a fundamental trade-off between discriminability and transferability in GNDEs and are supported by our numerical examples.

## 1 Introduction

Graph Neural Differential Equations (GNDEs) represent an innovative extension of Graph Neural Networks (GNNs), where the forward pass is formulated as the solution of an ordinary differential equation (ODE), with the derivative function parameterized by a GNN. Introduced by Poli et al. (2019), this framework generalizes Neural ODEs Chen et al. (2018) to the graph domain, enhancing sample efficiency and generalization performance on networked data compared to traditional discrete GNNs. GNDEs and their variants are particularly effective in modeling continuous-time phenomena in networked dynamical systems and often outperform conventional deep learning models that do not explicitly account for graph structures Poli et al. (2019); Choi et al. (2022); Xu et al. (2023); Chen et al. (2024).

Similar to Neural ODEs, GNDEs can be trained using standard backpropagation and the adjoint sensitivity method, which offers memory efficiency. However, training GNDEs on large, dense graphs remains a challenging problem. In GNNs, graph convolutions employ shared coefficients across nodes, allowing generalization across different graphs. GNDEs share this capability, prompting an important question: Can GNDEs trained on moderately sized graphs be transferred to larger, structurally similar graphs while preserving high prediction accuracy?

In discrete settings, the transferability of Graph Neural Networks (GNNs) across graphs of varying sizes has garnered significant attention. Recent advances have introduced Graphon Neural Networks (Graphon-NNs) as limit objects of GNNs, establishing theoretical bounds on the

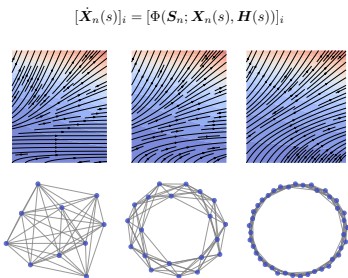

$$[\dot{\boldsymbol{X}}_n(s)]_i = [\Phi(\boldsymbol{S}_n; \boldsymbol{X}_n(s), \boldsymbol{H}(s))]_i$$

Figure 1: Vector fields generated by the GNDE model over cyclic kNN (k=4) graphs with $N = 10, 20, 40$ nodes.

approximation error between GNNs and their corresponding Graphon-NNs. These results reveal a fundamental trade-off between discriminability and transferability. However, directly extending these bounds to Graph Neural Differential Equations (GNDEs) is challenging. While GNDEs can be viewed as continuous-depth limits of GNNs, the process of taking such limits causes the existing bounds to diverge. This divergence arises from the infinite-dimensional nature of continuous-depth models, which requires a fundamentally different analytical framework.

To address this issue, we leverage tools from theory of dynamical systems and graph limits to develop a new framework for analyzing the transferability of GNDEs, ensuring that meaningful bounds hold in the continuous-depth setting. We introduce Graphon Neural Differential Equations (Graphon-NDEs) as continuous counterparts of GNDEs and derive rigorous results that capture the complex trade-offs between discriminability and transferability. Our framework tackles the unique challenges posed by the continuous-depth nature of GNDEs, such as handling infinite-dimensional dynamics and maintaining stability over continuous evolutions—issues that are absent in discrete GNNs. By overcoming these obstacles, our findings provide new insights into the scalability and generalization of GNDEs, laying a theoretical foundation for the development of robust and flexible graph-based neural differential equation models capable of adapting to complex and diverse graph structures.

## RELATED WORK

Graphons and their associated convergent graph sequences have been foundational in mathematics Borgs et al. (2008; 2012); Lovász (2012) and have gained significant traction in machine learning due to their ability to represent large-scale networks as limit objects.

In networked dynamical systems, graphons enable the transition from finite network models to their infinite or continuum counterparts. By deriving limits of large network dynamics, researchers gain insights into complex phenomena such as *chimera states* Abrams & Strogatz (2004); Kuramoto & Battogtokh (2002), *multistability* Girnyk et al. (2012); Wiley et al. (2006), and *synchronization*. However, rigorously justifying these limits and ensuring that continuum models accurately represent the dynamics of the original systems poses significant challenges, often requiring non-trivial, problem-specific efforts and is a very active line of research in applied math Paul & Trélat (2022).

Building on recent advancements in deriving limits for nonlinear evolution equations on graphs Medvedev (2014); Paul & Trélat (2022) and transferability in Graph Convolutional Networks (GCNs) using the graphon framework Maskey et al. (2023), we present new transferability results for Graph Convolutional Neural Differential Equations (GCNDEs). Specifically, we address both complete weighted graphs sampled from a *Lipschitz graphon* Ruiz et al. (2020) and simple graphs generated from $\{0, 1\}$-valued graphons Ruiz et al. (2021a;b); Morency & Leus (2021). Using functional calculus techniques inspired by Maskey et al. (2023) and generalizing them to the continuous-depth setting through the introduction of new norms and stability theory of dynamical systems, we establish explicit convergence rates of $\mathcal{O}(1/n)$ for Lipschitz graphons and $\mathcal{O}(1/n^c)$ for $\{0, 1\}$-valued graphons, where $c$ depends on the box-counting dimension of the boundary of the support.

To maintain simplicity while preserving the core ideas of our approach, we adopt a standard GCN architecture. However, our framework is flexible and can naturally extend to GNDEs with Lipschitz filters as introduced in Maskey et al. (2023). In future work, we plan to explore further generalizations, including random generative models, to broaden the applicability of our method.

## 2 NOTATION AND PRELIMINARY CONCEPTS

For a positive integer $n$, we let $[n] := \{1, 2, \ldots, n\}$ and $\mathbb{Z}_n := \{0, 1, \ldots, n - 1\}$. We denote the unit interval as $I := [0, 1]$ and $I^2 := I \times I$. For an interval $J \subseteq I$, by $|J|$ we denote the length of $J$, and we define the indicator function $\chi_J : J \to \{0, 1\}$ as

$$\chi_J(u) := \begin{cases} 1, & \text{if } u \in J, \\ 0, & \text{otherwise.} \end{cases}$$

The function space $L^p(I; \mathbb{R}^{1 \times F})$ consists of all $L^p$-integrable vector valued functions mapping $I$ to $\mathbb{R}^{1 \times F}$, where $1 \leq p \leq \infty$ and $F$ denotes the number of features. For a vector-valued function

$\mathbf{G} = [G_f : f \in [F]] \in L^\infty(I; \mathbb{R}^{1 \times F})$, we define the integral of $\mathbf{G}$ on $I$ as

$$\int_I \mathbf{G}(u)du := \left[ \int_I G_f(u)du : f \in [F] \right].$$

We need a space for time-dependent functions. For $\Omega \subset [0, \infty)$ and $1 \le p \le \infty$, the space $C^1(\Omega; L^p(I; \mathbb{R}^{1 \times F}))$ is composed of functions $\mathbf{X} : I \times \Omega \to \mathbb{R}^{1 \times F}$ satisfying

1. For each $t \in \Omega$, $\mathbf{X}(\cdot, t)$ belongs to $L^p(I; \mathbb{R}^{1 \times F})$.
2. For each $u \in I$, $\mathbf{X}(u, \cdot)$ is continuously differentiable.

We remark that $u \in I$ and $t \in \Omega$ represents the spatial and time variables, respectively, in the function $\mathbf{X}$.

Consider an undirected, simple graph $\mathcal{G}_n = \langle V_n(\mathcal{G}_n), E_n(\mathcal{G}_n) \rangle$, where $V_n(\mathcal{G}_n)$ is the set of nodes with cardinality $|V_n(\mathcal{G}_n)| = n$, and $E_n(\mathcal{G}_n) \subseteq V_n(\mathcal{G}_n) \times V_n(\mathcal{G}_n)$ represents the set of edges. The weight adjacency matrix is denoted as $\boldsymbol{W}_{\mathcal{G}_n} : E_n(\mathcal{G}_n) \to [0, 1]$, where $[\boldsymbol{W}_{\mathcal{G}_n}]_{ij} = [\boldsymbol{W}_{\mathcal{G}_n}]_{ji} \neq 0$ if $(i, j) \in E_n(\mathcal{G}_n)$. A *graph feature matrix* $\boldsymbol{X}_n \in \mathbb{R}^{n \times F}$ assigns a feature vector $[\boldsymbol{X}_n]_{i,:} \in \mathbb{R}^{1 \times F}$ to each node $i \in V_n(\mathcal{G}_n)$.

## 2.1 GRAPH NEURAL NETWORKS AND GRAPH NEURAL DIFFERENTIAL EQUATIONS

Graph Neural Networks (GNNs), introduced by Scarselli et al. (2008), are foundational tools for learning from graph-structured data. *Graph Neural Differential Equations (GNDEs)* extend GNNs to continuous-time systems, modeling the evolution of node features as

$$\begin{cases} \dot{\boldsymbol{X}}_n(s) = \mathbf{F}_{\mathcal{G}_n}(s; \boldsymbol{X}_n(s); \theta(s)), \\ \boldsymbol{X}_n(0) = \boldsymbol{G}_n, \end{cases} \tag{1}$$

where $\boldsymbol{X}_n(s) \in \mathbb{R}^{n \times F}$ represents the node features at time $s$ with initial node feature matrix $\boldsymbol{G}_n \in \mathbb{R}^{n \times F}$, and $\mathbf{F}_{\mathcal{G}_n}$ is a GNN parameterized by trainable functions $\theta(s)$. GNDEs address the limitations of fixed-depth GNNs, enabling them to capture complex temporal and structural patterns, particularly in dynamic graph scenarios.

A distinctive feature of GNDEs is that their parameters, $\theta(s)$, are graph-agnostic, allowing for potential transferability across different graph structures. This raises a key question: *Under what conditions can GNDEs trained on one graph be effectively transferred to another?* Addressing this question is crucial for leveraging models trained on moderately-sized graphs to perform well on larger, yet structurally similar, graphs, and therefore alleviating the computational challenging for large scale graphs.

We focus on Graph Convolutional Neural Differential Equations (GCNDEs) as a specific instance of GNDEs. Our aim is to establish a rigorous mathematical framework for understanding and improving the transferability of these models. While we concentrate on this particular architecture, the principles and techniques we develop are expected to extend to broader GNDE architectures, which we leave for future research.

**Graph Convolutional Networks (GCNs) and Graph Convolutional Neural Differential Equations (GCNDEs)** One of the most prevalent GNN architectures is the GCNs, introduced by Bruna et al. (2013) and popularized by Kipf & Welling (2016). GCNs excel at capturing local graph structures by extending traditional convolutions to graphs using spectral methods, effectively aggregating information from neighboring nodes. This capability makes GCNs particularly successful in tasks such as node classification and link prediction.

We adopt the formulation of GCNs using a *graph shift operator (GSO) $\boldsymbol{S}_n$*, which generalizes prior approaches by encompassing them as special cases Ruiz et al. (2020). A GSO $\boldsymbol{S}_n$ is a matrix that encodes the graph structure, where $[\boldsymbol{S}_n]_{ij} = [\boldsymbol{S}_n]_{ji} \neq 0$ if $i = j$ or $(i, j) \in E_n(\mathcal{G}_n)$. Common choices for $\boldsymbol{S}_n$ include the (weighted) adjacency matrix or the graph Laplacian. The graph convolution operation for a graph signal $\boldsymbol{x} \in \mathbb{R}^n$ is defined by

$$\boldsymbol{h} *_{\boldsymbol{S}_n} \boldsymbol{x} := \sum_{k=0}^{K-1} h_k \boldsymbol{S}_n^k \boldsymbol{x} = \mathbf{h}(\boldsymbol{S}_n)\boldsymbol{x},$$

where $\mathbf{h}(x) := \sum_{k=0}^{K-1} h_k x^k$, $x \in \mathbb{R}$, is a polynomial determined by the filter $\boldsymbol{h} = [h_k : k \in \mathbb{Z}_K]$. With the notation of graph convolutional operation, the output of $\ell$-th layer of a GCN is given by

$$[\boldsymbol{X}_{n,\ell}]_{:,f} := \rho \left( \sum_{g=1}^{F} \mathbf{h}_\ell^{fg} *_{\boldsymbol{S}_n} [\boldsymbol{X}_{n,\ell-1}]_{:,g} \right), \quad f \in [F], \; \ell \in [L], \tag{2}$$

where $\boldsymbol{X}_{n,0} := \boldsymbol{G}_n \in \mathbb{R}^{n \times F}$ is the input feature matrix and $\rho$ is a nonlinear activation function. The output of this $L$-layer GCN can be compactly represented as

$$\boldsymbol{X}_{n,L} := \Phi(\boldsymbol{S}_n; \boldsymbol{X}_{n,0}; \boldsymbol{H}), \tag{3}$$

where the tensor $\boldsymbol{H}_{f,g,\ell,:} := \mathbf{h}_\ell^{fg} \in \mathbb{R}^K$ contains the trainable filter coefficients for all layers.

*Graph Convolutional Neural Differential Equations (GCNDEs)* further refine GNDEs by evolving node features according to

$$\begin{cases} \dot{\boldsymbol{X}}_n(s) = \Phi(\boldsymbol{S}_n; \boldsymbol{X}_n(s); \boldsymbol{H}(s)), \\ \boldsymbol{X}_n(0) = \boldsymbol{G}_n \in \mathbb{R}^{n \times F}, \end{cases} \tag{4}$$

where the parameters in $\boldsymbol{H}(s) \in \mathbb{R}^{F \times F \times L \times K}$ are independent of the graph size $n$. This independence enables the model to be transferred to new graphs by only adjusting $\boldsymbol{S}_n$ once trained.

## 2.2 GRAPHON AS GRAPH LIMITS AND GRAPHON CONVOLUTIONAL NEURAL NETWORKS

Recent advances in the transferability of GCNs utilize graphons to model the continuous limit of large graphs. To build on this, we first review the key theoretical concepts that form the foundation of our approach.

A *graphon* is a bounded, symmetric, and measurable function $\mathbf{W} : I^2 \to I$, serving as a continuous generalization of an adjacency matrix. In this setting, nodes $i$ and $j$ are represented by points $u_i, u_j \in I$, with the edge weight between them given by $\mathbf{W}(u_i, u_j)$.

To quantify the convergence of a graph sequence $\{\mathcal{G}_n\}$, we use *graph motifs* $\mathcal{F}$, which are arbitrary, unweighted, and undirected graphs. A *homomorphism* from $\mathcal{F} = \langle V(\mathcal{F}), E(\mathcal{F}) \rangle$ to a graph $\mathcal{G} = \langle V(\mathcal{G}), E(\mathcal{G}) \rangle$ is a mapping $\phi : V(\mathcal{F}) \to V(\mathcal{G})$ that preserves adjacency, meaning $(i,j) \in E(\mathcal{F})$ implies $(\phi(i), \phi(j)) \in E(\mathcal{G})$. The *homomorphism density* is defined as

$$t(\mathcal{F}, \mathcal{G}) := \frac{\hom(\mathcal{F}, \mathcal{G})}{|V(\mathcal{G})|^{|V(\mathcal{F})|}},$$

which measures the relative frequency of $\mathcal{F}$ appearing in $\mathcal{G}$ and quantifies structural similarity between graphs.

Homomorphisms from graphs to graphons are defined similarly to those between graphs. Let $t(\mathcal{F}, \mathbf{W})$ denote the density of homomorphisms from a graph $\mathcal{F}$ into a graphon $\mathbf{W}$. We say that a sequence of graphs $\{\mathcal{G}_n\}$ converges to the graphon $\mathbf{W}$ if, for all finite, unweighted, and undirected graphs $\mathcal{F}$,

$$\lim_{n \to \infty} t(\mathcal{F}, \mathcal{G}_n) = t(\mathcal{F}, \mathbf{W}).$$

Every graphon is the limit object of some convergent graph sequence, and, conversely, every convergent graph sequence converges to a unique graphon Lovász (2012). Thus, a graphon represents an entire class of graphs that, regardless of their size, belong to the same "graphon family." Graphons effectively capture the asymptotic behavior of dense graph sequences $\{\mathcal{G}_n\}$, providing a robust framework for analyzing large-scale networks. One can refer to more details in A.2.

**Graphon Convolutional Neural Networks (Graphon-CNNs)** Extending the concept of graph convergence, a sequence of GNNs can converge to a *graphon neural network*, which is a limit architecture defined by layers of graphon convolutions and nonlinear activations. In this framework, the continuous analogue of graph convolution operators, known as *graphon convolution operators*, is derived in Ruiz et al. (2020; 2021a). For a given graphon $\mathbf{W} : I^2 \to I$, the *graphon convolution operator*, denoted by $T_{\mathbf{W}}$, acting on a feature function $\mathbf{x} \in L^2(I; \mathbb{R})$ is defined as

$$T_{\mathbf{W}} \mathbf{x}(v) := \int_0^1 \mathbf{W}(u,v) \mathbf{x}(u) \, du, \quad v \in I.$$

This integral operator is self-adjoint and Hilbert-Schmidt, with eigenvalues lying on $[-1, 1]$ and accumulating around zero. For a filter $\boldsymbol{h} = [h_0, \ldots, h_{K-1}]^\top$, by introducing a polynomial $\mathbf{h}(x) := \sum_{k=0}^{K-1} h_k x^k$, $x \in \mathbb{R}$, the graphon convolution is defined by

$$\boldsymbol{h} *_{\mathbf{W}} \mathbf{x} := \sum_{k=0}^{K-1} h_k T_{\mathbf{W}}^k \mathbf{x} = \mathbf{h}(T_{\mathbf{W}})\mathbf{x},$$

where

$$T_{\mathbf{W}}^k \mathbf{x}(v) := \int_0^1 \mathbf{W}(u, v)(T_{\mathbf{W}}^{k-1}\mathbf{x})(u)\, du, \quad v \in I, \quad \text{and} \quad T_{\mathbf{W}}^0 := \boldsymbol{I},$$

where $\boldsymbol{I}$ is the identity operator. Let $\mathbf{X}_0 \in L^\infty(I; \mathbb{R}^{1 \times F})$ represent the input feature function. The $\ell$-th layer of a Graphon-CNN is given by:

$$[\mathbf{X}_\ell]_{:,f} = \rho\left( \sum_{g=1}^{F} \boldsymbol{h}_\ell^{fg} *_{\mathbf{W}} [\mathbf{X}_{\ell-1}]_{:,g} \right), \quad f \in [F],\ \ell \in [L], \tag{5}$$

where $\rho$ denotes a nonlinear activation function. The output is represented as:

$$\mathbf{X}_L := \Phi(\mathbf{W}; \mathbf{X}_0; \boldsymbol{H}),$$

where $\boldsymbol{H}$ contains the trainable filter parameters as those in 3.

## 3 GRAPHON CONVOLUTIONAL NEURAL DIFFERENTIAL EQUATIONS

Building on the transition from GCNs to Graphon Convolutional Neural Networks, we extend the concept of GCNDEs to the continuous domain by introducing *Graphon Convolutional Neural Differential Equations (Graphon-CNDEs)*. The system is defined by

$$\frac{\partial}{\partial t}\mathbf{X}(u, t) = \Phi(\mathbf{W}; \mathbf{X}(u, t); \mathbf{H}(t)),$$
$$\mathbf{X}(u, 0) = \mathbf{G}(u) \in L^\infty(I; \mathbb{R}^{1 \times F}), \tag{6}$$

where $\Phi$ denotes the graphon convolutional operator parameterized by

$$\mathbf{H}(t) = \left\{ [\boldsymbol{h}_\ell^{fg}]_k(t) : f, g \in [F], \ell \in [L], k \in \mathbb{Z}_K \right\},$$

and $\mathbf{W}$ represents the graphon characterizing the continuous structure of the underlying graph sequence. The initial graphon feature function $\mathbf{G}$ specifies the node features at time $t = 0$. For each $t > 0$, by $\mathbf{h}_{\ell,t}^{fg}$ we denote the polynomial determined by the filter $\boldsymbol{h}_\ell^{fg}(t)$, for all $f, g \in [F]$ and $\ell \in [L]$, that is

$$\mathbf{h}_{\ell,t}^{fg}(x) := \sum_{k=0}^{K-1} \left( [\boldsymbol{h}_\ell^{fg}]_k(t) \right) x^k, \quad x \in \mathbb{R}. \tag{7}$$

To ensure the validity of our newly introduced Graphon-CNDEs, it is crucial to establish their well-posedness. This guarantees the existence, uniqueness, and continuous dependence of solutions on the initial conditions. Notably, we achieve this under mild assumptions that require only measure-theoretic properties of the graphons, without imposing any topological regularity conditions. This flexibility allows our results to be applicable in very general spaces including generalizations to atom-free standard probability space.

- **AS0.** The convolutional filters are $A_0$-Lipschitz continuous about $t$, namely, for each $f, g \in [F], \ell \in [L], k \in \mathbb{Z}_K$, there holds $\left| [\boldsymbol{h}_\ell^{fg}]_k(t_1) - [\boldsymbol{h}_\ell^{fg}]_k(t_2) \right| \leq A_0 |t_1 - t_2|$, for all $t_1, t_2 \in \mathbb{R}$.

- **AS1.** The activation function $\rho$ is normalized Lipschitz, that is, $|\rho(x) - \rho(y)| \leq |x - y|$, $x, y \in \mathbb{R}$ and $\rho(0) = 0$.

**Theorem 3.1** (Well-posedness, proof in Section A.4). *Suppose that AS0 and AS1 hold. If $\mathbf{W} \in L^\infty(I^2; \mathbb{R})$ and $\mathbf{G} \in L^\infty(I; \mathbb{R}^{1 \times F})$, then for any $T > 0$, there exists a unique solution of IVP 6, such that $\mathbf{X} \in C^1([0, T]; L^\infty(I; \mathbb{R}^{1 \times F}))$.*

## 3.1 Graphon-CNDEs as Deterministic Generative Models for GCNDEs

By comparing the GCDE 4 and the Graphon-CNDE 6, we observe that both can share the same set of parameters **H**. For graphs derived from a common graphon family, this implies that GCDEs can be viewed as specific instances of Graphon-CNDEs. Consequently, Graphon-CNDEs serve as deterministic generative models for GCNDEs, offering insights into the structural properties of these networks.

We focus on two deterministic families of discrete convergent (as detailed in SubSection 2.2) graph models, **Model I** for complete weighted graphs and **Model II** for simple unweighted graphs, constructed from a graphon **W**. Although deterministic, these models provide a foundation for understanding more complex random graph models and their behavior in machine learning tasks such as node classification, link prediction, and graph signal processing.

We split the unit interval $I$ into $n$ subintervals by setting $u_i := \frac{i-1}{n}$ and $I_i^{(n)} := [u_i, u_{i+1})$ for $i \in [n]$. In the following, we introduce different ways to model a sequence of graphs $\{\mathcal{G}_n\}_{n \in \mathbb{N}}$ with node features generated from the graphon **W** and a graphon feature function $\mathbf{G} \in L^2(I; \mathbb{R}^{1 \times F})$.

**Model I (Complete Graphs with weighted adjacent matrix)**  Suppose that **W** is a graphon and **G** is a graphon feature function. For each $n \in \mathbb{N}$, a complete graph $\mathcal{G}_n$ on $n$ nodes is defined by

$$\mathcal{G}_n := \langle [n], [n] \times [n] \rangle,$$

where we construct the weighted adjacency matrix $\boldsymbol{W}_{\mathcal{G}_n} \in \mathbb{R}^{n \times n}$ by direct sampling on the graphon **W** over the mesh grid as

$$[\boldsymbol{W}_{\mathcal{G}_n}]_{ij} := \mathbf{W}(u_i, u_j), \quad i, j \in [n]. \tag{8}$$

This model is particularly useful in scenarios where a fully connected network structure is required, such as in dense communication networks or certain types of recommendation systems, where understanding interactions between all nodes is crucial.

Consistently, the corresponding node feature matrix $\boldsymbol{G}_n \in \mathbb{R}^{n \times n}$ on $\mathcal{G}_n$ is generated by sampling on the graphon feature function **G** as

$$[\boldsymbol{G}_n]_{i,:} := \mathbf{G}(u_i), \quad i \in [n]. \tag{9}$$

**Model II (Simple Graphs with binary adjacent matrix)**  Suppose that $\mathbf{W} : I^2 \to \{0, 1\}$ is a graphon for simple graphs having binary weights and **G** is a graphon feature function. We denote by $\mathbf{W}^+$ the support set of **W**, that is $\mathbf{W}^+ := \{(u, v) : \mathbf{W}(u, v) = 1\}$. For each $n \in \mathbb{N}$, we construct a simple graph $\mathcal{G}_n$ as

$$\mathcal{G}_n := \langle [n], E(\mathcal{G}_n) \rangle,$$

where the edge set $E(\mathcal{G}_n)$ is defined by

$$E(\mathcal{G}_n) := \{(i, j) \in [n] \times [n] : (I_i^{(n)} \times I_j^{(n)}) \cap \mathbf{W}^+ \neq \emptyset\},$$

and the adjacent matrix $\boldsymbol{W}_{\mathcal{G}_n}$ is defined by

$$[\boldsymbol{W}_{\mathcal{G}_n}]_{ij} := \begin{cases} 1, & \text{if } (i, j) \in E(\mathcal{G}_n), \\ 0, & \text{otherwise.} \end{cases} \tag{10}$$

We remark that here $[\boldsymbol{W}_{\mathcal{G}_n}]_{ij}$ represents the binary connectivity between nodes $i$ and $j$ of the graph $\mathcal{G}_n$. This model is well-suited for generating network structures with binary relations, which are prevalent in social networks, citation graphs, and biological networks. The corresponding node feature matrix $\boldsymbol{G}_n$ for graph $\mathcal{G}_n$ is defined, with the help of graphon feature function **G**, by

$$[\boldsymbol{G}_n]_{i,:} := \frac{1}{|I_i^{(n)}|} \int_{I_i^{(n)}} \mathbf{G}(u) \, du, \quad i \in [n]. \tag{11}$$

### 3.2 PROBLEM FORMULATION

Given a graph $\mathcal{G}_n$ constructed under either model I or II, we define GSO as $\boldsymbol{S}_n = \frac{1}{n}\boldsymbol{W}_{\mathcal{G}_n}$, where the scaling factor $\frac{1}{n}$ is necessary for convergence. Then the corresponding graph neural differential equation is formulated as

$$
\begin{aligned}
\frac{\partial}{\partial t}\boldsymbol{X}_n(t) &= \Phi(\boldsymbol{S}_n; \boldsymbol{X}_n(t); \mathbf{H}(t)), \\
\boldsymbol{X}_n(0) &= \boldsymbol{G}_n
\end{aligned}
\tag{12}
$$

where $\boldsymbol{G}_n$ is the corresponding discrete node features of $\mathbf{G}$ over $\mathcal{G}_n$(cf.9 or 11).

As $n$ increases, both the matrix $\boldsymbol{W}_{\mathcal{G}_n}$ and the initial condition $\mathbf{G}_n$ converge to the graphon $\mathbf{W}$ and the continuous initial condition $\mathbf{G}$, respectively. Consequently, we expect the solution $\boldsymbol{X}_n$ of the discrete model in 12 to converge towards the solution $\mathbf{X}$ of the continuum model in 6. The primary research questions we address are as follows:

(A) **Accuracy of Continuum Approximation:** Does the continuum model 6 provide an accurate approximation of the dynamics of the discrete model 12 for large finite $n$? If so, what is the precise sense in which the solutions of the integro-differential equation approximate those of the discrete model?

(B) **Applicability Across Network Topologies:** To what extent is the continuum limit applicable across various network topologies? Can it be generalized beyond specific structures like k-nearest-neighbor graphs on a ring, extending to more complex and realistic networks such as small-world or scale-free graphs?

Direct comparison between the continuous output $\mathbf{X}(\cdot, t)$, defined over the interval $I$, and the discrete output $\boldsymbol{X}_n(t)$, defined over the index set $[n]$, is challenging. To bridge this gap, we introduce a Graphon-NDE induced by a GNDE.

For $n \in \mathbb{N}$, with a discrete adjacency matrix $\boldsymbol{W}_{\mathcal{G}_n}$, and a discrete node feature matrix $\boldsymbol{G}_n$ being given, we define a graphon $\mathbf{W}_n : I^2 \to \mathbb{R}$ (corresponding to $\boldsymbol{W}_{\mathcal{G}_n}$) by

$$
\mathbf{W}_n(u, v) := \sum_{i,j \in [n]} [\boldsymbol{W}_{\mathcal{G}_n}]_{ij} \chi_{I_i^{(n)}}(u) \chi_{I_j^{(n)}}(v), \quad u, v \in I,
\tag{13}
$$

and a graphon feature function $\mathbf{G}_n : I \to \mathbb{R}$ (corresponding to $\boldsymbol{G}_n$) by

$$
\mathbf{G}_n(u) := \sum_{i \in [n]} [\boldsymbol{G}_n]_{i,:} \chi_{I_i^{(n)}}(u), \quad u \in I.
\tag{14}
$$

Then the induced Graphon-NDE is formulated as

$$
\begin{aligned}
\frac{\partial}{\partial t}\mathbf{X}_n(u, t) &= \Phi(\mathbf{W}_n; \mathbf{X}_n(u, t); \mathbf{H}(t)), \\
\mathbf{X}_n(u, 0) &= \mathbf{G}_n(u),
\end{aligned}
\tag{15}
$$

where $\Phi(\mathbf{W}_n; \mathbf{X}_n(u, t); \mathbf{H}(t))$ denotes the graphon convolutional neural network parameterized by $\mathbf{H}(t)$ over the induced graphon $\mathbf{W}_n$, and $\mathbf{G}_n$ is the graphon representation of initial node feature matrix $\boldsymbol{G}_n$. This construction embeds the dynamics of the discrete graph domain 12 "equivalently" into the continuous graphon domain 15, and enables a systematic comparison between GNDEs and their continuous counterparts, facilitating the analysis of their convergence and scalability.

We proceed to consider the quality of approximation of the solution of 15 to the solution of 6 as $n \to \infty$, for both Models I and II. We list some mild assumptions, partially used for Models I and II, on convolutional filters $\mathbf{H}$, graphon $\mathbf{W}$ and initial graphon feature function $\mathbf{G}$.

- **AS2.** For each $t \geq 0$, $f, g \in [F]$ and $\ell \in [L]$, the function $\mathbf{h}_{\ell,t}^{fg}$ is Lipschitz continuous with Lipschitz constant $\mathrm{Lip}(\mathbf{h}_{\ell,t}^{fg}(\cdot))$.

- **AS3.** The graphon $\mathbf{W}$ is $A_1$-Lipschitz, that is, $|\mathbf{W}(u_2, v_2) - \mathbf{W}(u_1, v_1)| \leq A_1(|u_2 - u_1| + |v_2 - v_1|)$, for all $v_1, v_2, u_1, u_2 \in I$.

- **AS4.** The initial graphon feature function $\mathbf{G} = [G_f : f \in [F]] \in L^\infty(I; \mathbb{R}^{1 \times F})$ is $A_2$-Lipschitz, that is, for each $f \in [K]$, $|G_f(u_2) - G_f(u_1)| \leq A_2|u_2 - u_1|$, for all $u_1, u_2 \in I$.

Below, we present the linear convergence rate of solutions for Model I.

**Theorem 3.2** (proof in A.6). *Suppose that AS0-AS4 hold. Let $\mathbf{W}_n$ and $\mathbf{G}_n$ be defined by 13 and 14 with coefficients 8 and 9, respectively. Let $\mathbf{X}$ and $\mathbf{X}_n$ denote the solutions of 6 and 15, respectively. Then it holds that*

$$\|\mathbf{X} - \mathbf{X}_n\|_{C([0,T];L^2(I;\mathbb{R}^{1 \times F}))} \leq \frac{C}{n}, \tag{16}$$

*where $C$ is constant independent of $n$ and only depends on $\mathbf{H}$, $\mathbf{G}$, $\mathbf{W}$, $A_1$, $A_2$ and $T$ with explicit formula provided in 56.*

The proof of Theorem 3.2 requires only Lipschitz continuity on filters, unlike the highly regular filters in Ruiz et al. (2020); Keriven et al. (2020), and improves the convergence rate from $\mathcal{O}(1/\sqrt{n})$ in Ruiz et al. (2020) to $\mathcal{O}(1/n)$. This result also extends Theorem 5.4 of Maskey et al. (2023) to GNDEs with a continuous-depth setting.

We mention that the graphons for simple graphs are discontinuous, so in general, AS1 is not satisfied for Model II. We tackle this problem by employing the concept of upper box-counting dimension to characterize the complexity of the boundary of $\mathbf{W}^+$.

**Theorem 3.3** (proof in A.6). *Suppose that AS0-AS2 and AS4 hold. Let $\mathbf{W} : I^2 \to \{0, 1\}$ be a graphon for simple graphs with $b := \overline{\dim}_{\mathrm{B}}(\partial \mathbf{W}^+) \in [1, 2)$. Let $\mathbf{W}_n$ and $\mathbf{G}_n$ be defined by 13 and 14 with coefficients 10 and 11, respectively. Let $\mathbf{X}$ and $\mathbf{X}_n$ denote the solutions of 6 and 15, respectively. Then for any $\epsilon > 0$, there exists a positive integer $N_{\mathbf{W}}$ (depending on $\mathbf{W}$) such that when $n > N_{\mathbf{W}}$, it holds that*

$$\|\mathbf{X} - \mathbf{X}_n\|_{C([0,T];L^2(I;\mathbb{R}^{1 \times F}))} \leq \frac{\widetilde{C}}{n^{1 - \frac{b+\epsilon}{2}}}, \tag{17}$$

*where $\widetilde{C}$ is a constant independent of $n$, and only depends on $\mathbf{H}$, $\mathbf{G}$, $\mathbf{W}$, $A_2$ and $T$ with explicit formula provided in 64.*

Theorem 3.3 deals with irregular graphons that are merely measurable, offering convergence rates not presented in prior work Ruiz et al. (2021b;a); Morency & Leus (2021); Maskey et al. (2023).

In summary, Theorems 3.2 and 3.3 address problem (A) in Section 3.2 by providing explicit approximation bounds within the relevant function space, and tackle problem (B) by imposing purely measure-theoretic or mild regularity assumptions, thereby covering a broad family of graphs. As a byproduct, one can derive transferability bounds for GNDEs from our main theorems using the triangle inequality: for graphs of sizes $n_1$ and $n_2$ sampled as in Model I, the bound is $\mathcal{O}(\frac{1}{n_1} + \frac{1}{n_2})$, and for Model II, $\mathcal{O}(\frac{1}{n_1^c} + \frac{1}{n_2^c})$ (where $c := 1 - \frac{b+\epsilon}{2}$ as in 17). Notably, for Model I, existing bounds for discrete GCNs with $\tilde{L}$ layers behave as $\mathcal{O}(\frac{C_{\tilde{L}}}{n})$, with $C_{\tilde{L}} \to \infty$ as $\tilde{L} \to \infty$, highlighting the need for a new analytic framework for GCNDEs, even though they can be viewed as continuous limits of residual GCNs.

## 4 NUMERICAL RESULTS

In the following sections, we illustrate the GNDEs transferability results via several examples with additional set-up and computation details provided in Appendix A.1

**Transfer Learning of Nonlinear Heat Equations on Complete Weighted Graphs** We investigate the transferability of GNDEs in modeling nonlinear heat equation dynamics Medvedev (2014) across graphs of varying sizes. We train GNDE models on graphs $\mathcal{G}_n$ of different sizes, with $n \in \{20, 40, 60, 80, 100\}$ nodes, all sampled from the same underlying graphon. The learned parameters are transferred to predict the same type of dynamics on a larger graph with $N = 500$ nodes. The GNDE, parameterized by a GCN ($L = 2, K = 2, F = 1$) is trained using an MSE loss function and optimized with ADAM (lr $= 0.001$, $\beta_1 = 0.9$, $\beta_2 = 0.999$) based on a single training trajectory of the nonlinear heat equation defined over $\mathcal{G}_n$ (see equation 18).

We evaluate transferability by measuring the test $\ell^2$ error between the true dynamics $\boldsymbol{Y}_N(1)$ and the GNDE-predicted dynamics $\boldsymbol{X}_{N,n}(1)$ on $\mathcal{G}_N$, where the parameters were learned on $\mathcal{G}_n$. The relative error $\frac{\|\boldsymbol{Y}_N(1) - \boldsymbol{X}_{N,n}(1)\|_2}{\|\boldsymbol{Y}_N(1)\|_2}$ is shown in Figure 3 (a). The errors are small, on the order of $\mathcal{O}(10^{-2})$, and exhibit a general decay with increasing graph sizes. The fluctuations in error can be attributed to training challenges for larger number of nodes, but given the two-digit accuracy of the theoretical bound ($\mathcal{O}(1/500)$), these variations are expected. These findings demonstrate that GNDEs can learn complex physical dynamics on smaller graphs (even with $n = 20$) and effectively transfer this knowledge to larger systems ($N = 500$, see Figure 2), enabling scalable modeling of physical processes on graphs.

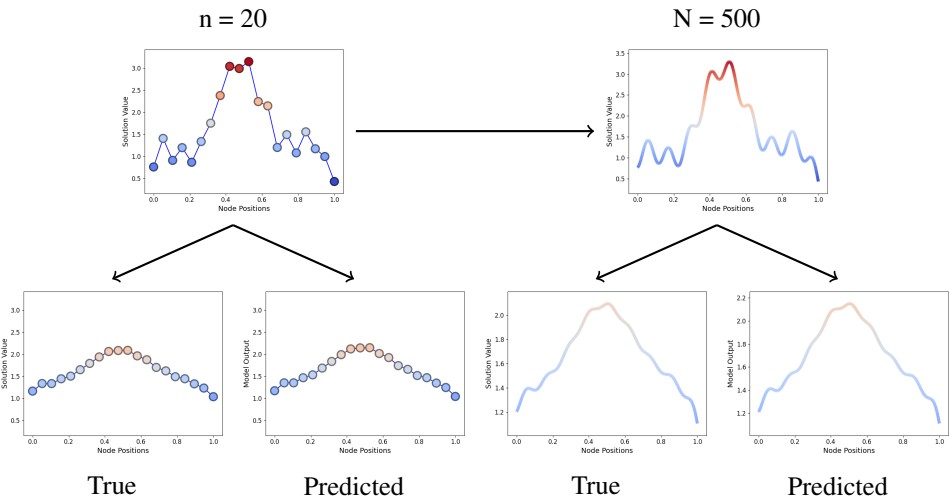

Figure 2: Top: Initial conditions for the nonlinear heat equation with $n = 20$ and $N = 500$ nodes, respectively. Bottom: True and trained GNDE prediction comparison of heat distribution at $t = 1$ for the given graph. Bottom left model is transferred to bottom right to make $N = 500$ prediction.

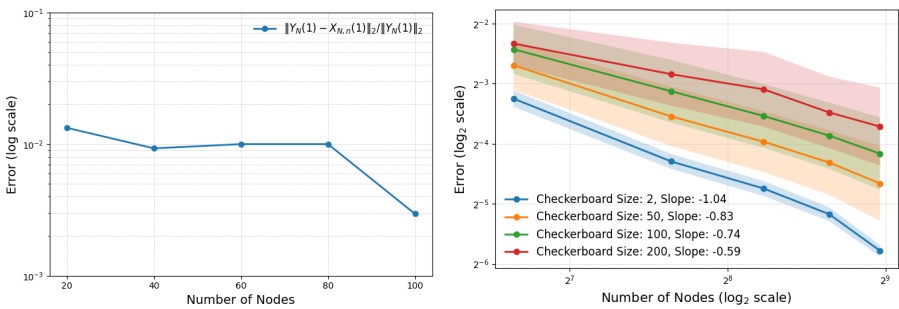

Figure 3: **Left:** GNDE relative prediction errors for nonlinear heat equations on a complete graph with 500 nodes. **Right**: Log-log convergence plot for transferability bounds of Checkerboard graphons.

**Transferability of GCNDEs on $\{0, 1\}$-valued Checkerboard Graphons with Varying Checkerboard Sizes** Theorem 3.3 demonstrates that the convergence rate of GNDEs is influenced by the box-counting dimension of the boundary of the $\{0, 1\}$-valued graphon support. While this upper bound provides valuable insights, it may not be optimal, and the necessity of this condition remains an open question. To empirically investigate, we study convergence rates on subgraphs derived from checkerboard graphons, where increasing checkerboard sizes correspond to higher box-counting dimensions. We generate diverse, smooth initial feature vectors using random Fourier series. For each checkerboard size, experiments are conducted on subgraphs with varying node counts, generated

via Model II, ensuring consistent GDE parameter transfer from the original graph. The relative $\ell^2$ norm error between the GNDE outputs on the original and subgraphs is computed, and results are averaged over multiple trials to account for variability in initial conditions. At $t = 0$, the expected convergence rate of initial conditions sampled from subgraphs is $\mathcal{O}(1/n)$. As shown in the log-log convergence plots for the solution discrepancy at $t = 1$ ((see Figure 3 (b))), the error decay rate decreases with increasing box-counting dimensions, supporting our theoretical predictions and highlighting the influence of structural complexity on transferability.

**Transferability of GCNDEs on the Cora Citation Network**  We explore the transferability of GNDEs in node classification tasks. The Cora citation network is a benchmark dataset of 2708 scientific publications each described by 1433 features. Publications are classified into 7 categories. We use the standard data split Kipf & Welling (2016) with 140 training, 500 validation, and 1000 testing nodes. We train GNDE models on random subgraphs of various sizes. We consider subgraphs with $10 - 50\%$ of the original nodes. For each subgraph, we create a corresponding model which consists of a linear head mapping to an input dimension of 16, a GNDE parameterized by a GCN ($L = 2, K = 2, F = 16$), and a linear readout layer mapping to the 7 class labels for the final classification task. We train with cross-entropy loss using ADAM optimization (lr $= 0.001$, $\beta_1 = 0.9$, $\beta_2 = 0.999$) on the subgraph. Training takes place over 1000 epochs. To avoid overfitting, representative models were selected as the lowest validation loss model after an initial convergence period of 200 epochs. Transfer then occurs to the full graph and test set accuracy for the full dataset is recorded. Ten random subgraphs were tested for each proportion of nodes, and the average results are reported in Table 1. As we train on a larger proportion of nodes, we gain accuracy on full graph prediction.

Table 1: Cora Average Test Accuracies

| Nodes in Subgraph | 10% | 20% | 30% | 40% | 50% |
|---|---|---|---|---|---|
| Subgraph Accuracy | $25.0 \pm 5.7$ | $31.3 \pm 5.4$ | $30.5 \pm 4.6$ | $31.5 \pm 3.4$ | $33.8 \pm 3.3$ |
| Full Graph Accuracy | $26.5 \pm 7.4$ | $29.1 \pm 6.0$ | $30.0 \pm 4.5$ | $31.5 \pm 3.1$ | $34.0 \pm 2.3$ |

## CONCLUSION AND FUTURE WORK

Transferability involves defining models that generate graphs and establishing metrics to measure discrepancies when a fixed GNN is applied to graphs of different sizes, which can be viewed as a notion of generalization. We introduced Graphon-NDEs as a novel framework integrating dynamical systems theory with the study of transferability in GNNs to analyze transferability of GNDEs. Our approach bridges these fields, enabling Graphon-NDEs to serve as generative models for GNDEs while providing theoretical guarantees on approximation accuracy. The framework can also incorporate other approaches analyzing the transferability of GCNs, such as the *stability method* Gama et al. (2020); Kenlay et al. (2021a;b) and *sampling techniques* Keriven et al. (2020); Levie et al. (2021), each offering unique perspectives. While we focused on deterministic graphs, this work lays the foundation for future research on stochastic graphs and more complex network structures, enhancing the robustness and generalization of GNDEs across diverse settings. We believe our framework provides a promising starting point for future work integrating dynamical systems and machine learning.

## REPRODUCIBILITY STATEMENT

Reproducible code for all experiments will be fully open-sourced upon acceptance.

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
