## A  Supplemental Materials

### A.1  Numerical Examples

We provide additional details about the experiments we conducted.

**Transfer Learning of Nonlinear Heat Equations on Complete Weighted Graphs**   In this set of experiments, we utilize the torchdiffeq library Chen et al. (2018) with an adaptive Dormand-Price solver Dormand & Prince (1980), and extend the code implementation from Poli et al. (2019). We consider the graphon function $\mathbf{W}(x, y) = \exp\left(-\frac{(x-y)^2}{2\sigma^2}\right)$ with $\sigma = 0.15$ to construct graphs $\mathcal{G}_n$ with $n$ nodes as described in Model 1 in Section 8. For training, we use five graphs with $n \in \{20, 40, 60, 80, 100\}$ nodes and simulate the nonlinear heat equation on $\mathcal{G}_n$ given by:

$$\frac{d}{dt}u_i(t) = \frac{8}{n}\sum_{j=1}^{n}[W_{\mathcal{G}_n}]_{ij}\sin(u_j(t) - u_i(t)), \quad i \in [n]. \tag{18}$$

Table 2: Initial Conditions for Nonlinear Heat Equation

| Dataset | Initial Condition $\mathbf{F}(u)$ |
|---|---|
| Training Data | $\frac{1}{1+\exp\left(-10\left(u-\frac{1}{2}\right)\right)} + \frac{3}{10}\sin(10u) + \frac{1}{10}\sin(40u)$ |
| Validation Data | $2u + \frac{1}{2}\tanh(5u) + \frac{1}{5}\sin(25u)$ |
| Testing Data | $3\exp\left(-20\left(u-\frac{1}{2}\right)^2\right) + \exp\left(-30\left(u-0.1\right)^2\right)$ 
 $+ \exp\left(-50\left(u-0.9\right)^2\right) + \frac{1}{2}\sin(35u)\sin(20u)$ |

We simulate the true dynamics over 50 time points within $t \in [0, 1]$ for three distinct initial conditions, forming our training, validation, and testing split (see Table 2). These datasets consist of pairs of observations one time step apart from the respective dynamics. For each $n$, we construct a GNDE parameterized by a GCN with $L = 2, K = 2, F = 1$ and a hidden dimension of 16. We train using the ADAM optimizer over 500 epochs with MSE loss between the true training dynamics and predicted next timestep. We utilize early stopping criteria provided by validation error measurement, which halts training after convergence but before overfitting occurs. Once trained, we transfer the weights of the GNDEs trained on $\mathcal{G}_n$ to a GNDE on the much larger $\mathcal{G}_N$ graph with $N = 500$. These larger GNDEs are evaluated on next-step dynamics prediction on the test set dynamics on $\mathcal{G}_N$. Convergence is then assessed through calculation of the relative error $\frac{\|\mathbf{Y}_N(1) - \mathbf{X}_{N,n}(1)\|_2}{\|\mathbf{Y}_N(1)\|_2}$.

**Transferability of GCNDEs on Checkerboard Graphons with Varying Checkerboard Sizes** We evaluate the transferability of Graph Convolutional Neural Differential Equations (GCNDEs) sampled from checkerboard graphons with varying checkerboard sizes ranging from 2 to 200, whose box-counting dimension falls in (1,2). Checkerboard graphons are $\{0, 1\}$-valued and represent a structured, piecewise-constant connectivity pattern (see Figure 4). Using Model II, we generate simple graphs from these graphons and apply fixed GNDE parameters with a two-layer GCN ($F = 2, K = 2$) to compute model outputs at $t = 1$ without any additional training. The initial conditions are random Fourier polynomials of degree $D = 10$, defined as:

$$\mathbf{G}(u) = \sum_{k=1}^{D} a_k \cos(2\pi k u) + b_k \sin(2\pi k u),$$

where $a_k$ and $b_k$ are independently sampled from a uniform distribution, creating diverse and smooth signals over graph nodes. We conduct 10 trials for each configuration and report the mean and standard deviation of the errors. To approximate the graphon solution $\mathbf{X}$, we use a reference graph with $N_{\text{largest}} = 3000$ nodes, ensuring a robust evaluation of model transferability across different

graph sizes. We presented the log-log convergence plot of $\frac{\|\boldsymbol{X}_n(1) - \boldsymbol{X}_{3000}(1)\|_2}{\|\boldsymbol{X}_{3000}(1)\|_2}$ for number of nodes $n$ ranging in 100:100:500.

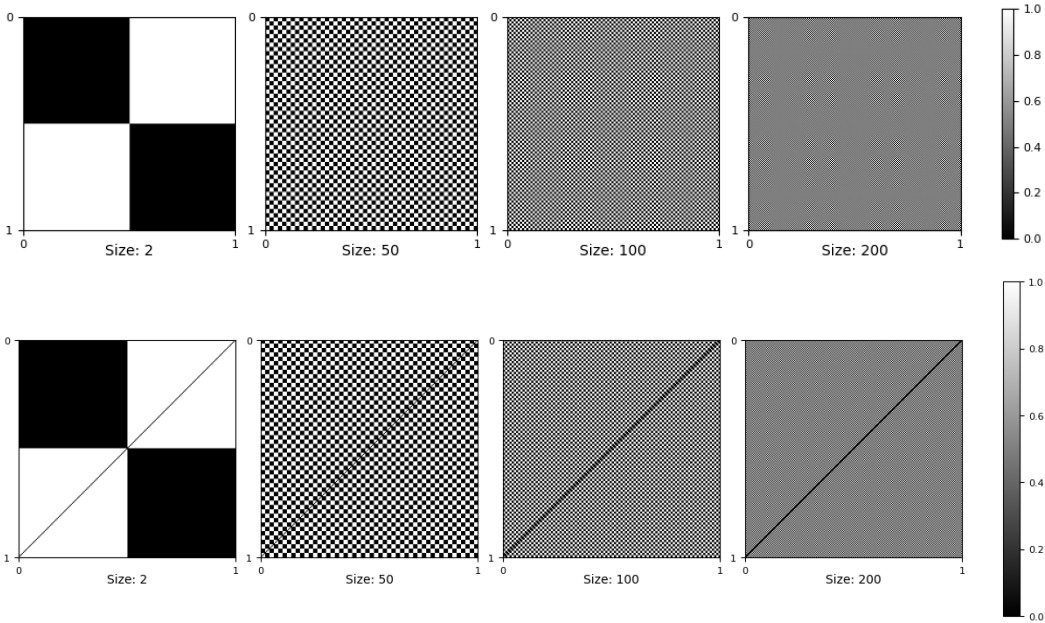

Figure 4: **Top**: Checkerboard graphons with sizes 2, 50, 100, and 200. **Bottom**: Piecewise constant approximations of checkerboard graphons using the adjacency matrix of graphs with 500 nodes sampled from the respective graphons.

## A.2 GRAPH LIMITS

To measure how similar two graphons are, we use something called the *cut-norm*. For any graphon **W**, its cut-norm is given by:

$$\|\mathbf{W}\|_\square := \sup_{S,T \subseteq I} \left| \int_{S \times T} \mathbf{W}(x, y) \, dx \, dy \right|, \tag{19}$$

where $S$ and $T$ are any subsets of $[0, 1]$. The cut-norm measures the maximum discrepancy in the graphon over any pair of subsets.

The *cut-distance* between two graphons **W** and **U** is defined as:

$$\delta_\square(\mathbf{U}, \mathbf{W}) := \inf_\phi \|\mathbf{U} - \mathbf{W}^\phi\|_\square, \tag{20}$$

where $\mathbf{W}^\phi(x, y) = \mathbf{W}(\phi(x), \phi(y))$ and $\phi$ is any measure-preserving mapping of $[0, 1]$. This ensures that the cut-distance is not affected by relabeling of the nodes, making it a robust way to compare graphons.

A graph sequence is convergent if and only if it is Cauchy with respect to the cut-distance. This means that the graphs in the sequence get closer and closer to each other in terms of structure. We define a *graph limit* as the equivalence class of graphons:

$$[\mathbf{W}] := \{\mathbf{U} \in \mathcal{W}_0 : \delta_\square(\mathbf{U}, \mathbf{W}) = 0\}. \tag{21}$$

In other words, two graphons are equivalent if they represent the same "limit" behavior.

We typically refer to both a graphon **W** and its equivalence class [**W**] simply as a "graphon". The set of all graphon equivalence classes, $\chi = \{[\mathbf{W}] : \mathbf{W} \in \mathcal{W}_0\}$, forms a compact metric space under the cut-distance $\delta_\square$ Lovász (2012). The sequence of graphs we constructed in Models I and II in Section 3 is convergent to graphon **W** Ruiz et al. (2020).

### A.3 FUNCTION SPACES

We define function spaces

$$L^\infty(I) := \left\{ G : I \to \mathbb{R} : \|G\|_{L^\infty(I)} := \mathrm{esssup}_{u \in I} |G(u)| < \infty \right\},$$

$$L^\infty(I; \mathbb{R}^{1 \times F}) := \left\{ \mathbf{G} : I \to \mathbb{R}^{1 \times F} : \|\mathbf{G}\|_{L^\infty(I; \mathbb{R}^{1 \times F})} := \mathrm{esssup}_{u \in I} \|\mathbf{G}(u)\|_2 < \infty \right\},$$

$$L^2(I) := \left\{ G : I \to \mathbb{R} : \|G\|_{L^2(I)} := \left( \int_I |G(u)|^2 du \right)^{\frac{1}{2}} < \infty \right\},$$

$$L^2(I; \mathbb{R}^{1 \times F}) := \left\{ \mathbf{G} : I \to \mathbb{R}^{1 \times F} : \|\mathbf{G}\|_{L^2(I; \mathbb{R}^{1 \times F})} := \left( \int_I \|\mathbf{G}(u)\|_2^2 du \right)^{\frac{1}{2}} < \infty \right\}.$$

We remark that for $\mathbf{G} = [G_f : f \in [F]] \in L^2(I; \mathbb{R}^{1 \times F})$, it is clear that

$$\|\mathbf{G}\|_{L^2(I; \mathbb{R}^{1 \times F})}^2 = \int_I \|\mathbf{G}(u)\|_2^2 du = \sum_{f \in [F]} \int_I |G_f(u)|^2 du = \sum_{f \in [F]} \|G_f\|_{L^2(I)}^2. \tag{22}$$

For $p \in [1, \infty]$, we define

$$C(\Omega; L^p(I; \mathbb{R}^{1 \times F})) := \{ \mathbf{X} : I \times \Omega \to \mathbb{R}^F : \text{for } \mathbf{X} = [X_f : f \in [F]], X_f(u, \cdot) \in C(\Omega) \text{ for each } u \in I,$$
$$\mathbf{X}(\cdot, t) \in L^p(I; \mathbb{R}^{1 \times F}) \text{ for each } t \in \Omega,$$
$$\|\mathbf{X}\|_{C(\Omega; L^p(I; \mathbb{R}^{1 \times F}))} := \sup_{t \in \Omega} \|\mathbf{X}(\cdot, t)\|_{L^p(I; \mathbb{R}^{1 \times F})} < \infty \}$$

and a subspace of $C(\Omega; L^p(I; \mathbb{R}^{1 \times F}))$ as

$$C^1(\Omega; L^p(I; \mathbb{R}^{1 \times F})) := \{ \mathbf{X} : I \times \Omega \to \mathbb{R}^F : \text{for } \mathbf{X} = [X_f : f \in [F]], X_f(u, \cdot) \in C^1(\Omega) \text{ for each } u \in I,$$
$$\mathbf{X}(\cdot, t) \in L^p(I; \mathbb{R}^{1 \times F}) \text{ for each } t \in \Omega \}.$$

### A.4 PROOF OF THEOREM 3.1

In this subsection, we provide the proof of Theorem 3.1. We recall that, for a convolutional filter $\boldsymbol{h} = [h_k : k \in \mathbb{Z}_K]$, by $\mathbf{h}$ we denote the polynomial determined by $\boldsymbol{h}$, that is, $\mathbf{h}(x) = \sum_{k \in \mathbb{Z}_K} h_k x^k$, $x \in \mathbb{R}$. By $|\mathbf{h}|$ we denote the polynomial $|\mathbf{h}|(x) := \sum_{k \in \mathbb{Z}_K} |h_k| x^k$, $x \in \mathbb{R}$. We proceed to present a helpful lemma to prove Theorem 3.1.

**Lemma A.1.** *Suppose that AS1 holds. Let $T > 0$ and $\mathbf{X}_1, \mathbf{X}_2 \in C([0, T]; L^\infty(I; \mathbb{R}^{1 \times F}))$. Then for all $t \in [0, T]$, it holds that*

$$\|\Phi(\mathbf{W}; \mathbf{X}_1(\cdot, t); \mathbf{H}(t)) - \Phi(\mathbf{W}; \mathbf{X}_2(\cdot, t); \mathbf{H}(t))\|_{L^\infty(I; \mathbb{R}^{1 \times F})} \le (F M_1)^L \|\mathbf{X}_1(\cdot, t) - \mathbf{X}_2(\cdot, t)\|_{L^\infty(I)}$$

*where*

$$M_1 := \max_{f, g \in [F], \ell \in [L], t \in [0, T]} \left| \mathbf{h}_{\ell, t}^{fg} \right| (\|\mathbf{W}\|_{L^\infty(I^2)}). \tag{23}$$

*Proof.* Let $t \in [0, T]$ be arbitrary but fixed. Let $\mathbf{X}_{0,t} := \mathbf{X}_1(\cdot, t)$ and $\mathbf{Y}_{0,t} := \mathbf{X}_2(\cdot, t)$. For each $f \in [F]$, we denote by $\mathbf{X}_{\ell,t} = [X_{\ell,t}^f : f \in [F]]$, $\mathbf{Y}_{\ell,t} = [Y_{\ell,t}^f : f \in [F]]$ the output of $\ell$-th layer in Graphon neural network with input feature functions $\mathbf{X}_{0,t}$ and $\mathbf{Y}_{0,t}$, respectively. It is clear that for each $\ell \in [L]$, we have $\mathbf{X}_{\ell,t}, \mathbf{Y}_{\ell,t} \in L^\infty(I; \mathbb{R}^{1 \times F})$. According to the updating formula 5 and AS1,

for each $f \in [F], \ell \in [L]$, we obtain that

$$
\begin{aligned}
\left\| X_{\ell,t}^f - Y_{\ell,t}^f \right\|_{L^\infty(I)} &= \left\| \rho \left( \sum_{g=1}^F \mathbf{h}_{\ell,t}^{fg}(T_\mathbf{W}) X_{\ell-1,t}^g \right) - \rho \left( \sum_{g=1}^F \mathbf{h}_{\ell,t}^{fg}(T_\mathbf{W}) Y_{\ell-1,t}^g \right) \right\|_{L^\infty(I)} \\
&\leq \left\| \sum_{g=1}^F \mathbf{h}_{\ell,t}^{fg}(T_\mathbf{W}) \left( X_{\ell-1,t}^g - Y_{\ell-1,t}^g \right) \right\|_{L^\infty(I)} \\
&\leq \sum_{g=1}^F \left\| \mathbf{h}_{\ell,t}^{fg}(T_\mathbf{W}) \right\|_{L^\infty(I) \to L^\infty(I)} \left\| X_{\ell-1,t}^g - Y_{\ell-1,t}^g \right\|_{L^\infty(I)} \\
&\leq \sqrt{\sum_{g=1}^F \left\| \mathbf{h}_{\ell,t}^{fg}(T_\mathbf{W}) \right\|_{L^\infty(I) \to L^\infty(I)}^2} \sqrt{\sum_{g=1}^F \left\| X_{\ell-1,t}^g - Y_{\ell-1,t}^g \right\|_{L^\infty(I)}^2} \\
&= \sqrt{\sum_{g=1}^F \left\| \mathbf{h}_{\ell,t}^{fg}(T_\mathbf{W}) \right\|_{L^\infty(I) \to L^\infty(I)}^2} \| \mathbf{X}_{\ell-1,t} - \mathbf{Y}_{\ell-1,t} \|_{L^\infty(I;\mathbb{R}^{1\times F})} \quad (24)
\end{aligned}
$$

Since $\mathbf{h}_{\ell,t}^{fg}$ is a polynomial for each $f, g \in [F], \ell \in [L]$, with noting $\|T_\mathbf{W}\|_{L^\infty(I) \to L^\infty(I)} \leq \|\mathbf{W}\|_{L^\infty(I^2)}$, it is clear that

$$
\left\| \mathbf{h}_{\ell,t}^{fg}(T_\mathbf{W}) \right\|_{L^\infty(I) \to L^\infty(I)} \leq |\mathbf{h}_{\ell,t}^{fg}|(\|\mathbf{W}\|_{L^\infty(I^2)}).
$$

Substituting the above inequality into 24, with the constant $M_1$ defined by 23, we obtain that

$$
\left\| X_{\ell,t}^f - Y_{\ell,t}^f \right\|_{L^\infty(I)} \leq \sqrt{F} M_1 \| \mathbf{X}_{\ell-1,t} - \mathbf{Y}_{\ell-1,t} \|_{L^\infty(I;\mathbb{R}^{1\times F})},
$$

which further implies

$$
\| \mathbf{X}_{\ell,t} - \mathbf{Y}_{\ell,t} \|_{L^\infty(I;\mathbb{R}^{1\times F})} = \sqrt{\sum_{f=1}^F \left\| X_{\ell,t}^f - Y_{\ell,t}^f \right\|_{L^\infty(I)}^2} \leq F M_1 \| \mathbf{X}_{\ell-1,t} - \mathbf{Y}_{\ell-1,t} \|_{L^\infty(I;\mathbb{R}^{1\times F})}.
$$

Using induction, we get

$$
\| \mathbf{X}_{L,t} - \mathbf{Y}_{L,t} \|_{L^\infty(I;\mathbb{R}^{1\times F})} \leq (F M_1)^L \| \mathbf{X}_{0,t} - \mathbf{Y}_{0,t} \|_{L^\infty(I)}.
$$

The proof is complete by noting that $\mathbf{X}_{L,t} = \Phi(\mathbf{W}; \mathbf{X}_1(\cdot,t); \mathbf{H}(t))$, $\mathbf{Y}_{L,t} = \Phi(\mathbf{W}; \mathbf{X}_2(\cdot,t); \mathbf{H}(t))$, $\mathbf{X}_{0,t} = \mathbf{X}_1(\cdot,t)$ and $\mathbf{Y}_{0,t} = \mathbf{X}_2(\cdot,t)$.

$\square$

*Proof of Theorem 3.1.* The proof is based on the Banach contraction mapping principles. We include the details for completeness. Let $T > 0$ be arbitrary but fixed, and $0 < \tau < \frac{1}{2(M_1 F)^L}$. We define a subspace $\mathcal{S}_\mathbf{G}$ of $C([0,\tau]; L^\infty(I;\mathbb{R}^{1\times F}))$, associated with $\tau$, by

$$
\mathcal{S}_\mathbf{G} := \left\{ \mathbf{X} : \mathbf{X} \in C([0,\tau]; L^\infty(I;\mathbb{R}^{1\times F})), \mathbf{X}(\cdot,0) = \mathbf{G} \right\}.
$$

Moreover, we define an integral operator $K : \mathcal{S}_\mathbf{G} \to \mathcal{S}_\mathbf{G}$ by

$$
[K\mathbf{X}](u,t) := \mathbf{G}(u) + \int_0^t \Phi(\mathbf{W}; \mathbf{X}(u,s); \mathbf{H}(s)) ds. \quad (25)
$$

It follows that we can rewrite the IVP for 6 as the fixed point equation $\mathbf{X} = K\mathbf{X}$. We show below that $K$ is a contraction. For any $\mathbf{X}_1, \mathbf{X}_2 \in \mathcal{S}_\mathbf{G}$, according to the definition of norm in $C([0,\tau]; L^\infty(I;\mathbb{R}^{1\times F}))$, we have

$$
\begin{aligned}
\| K\mathbf{X}_1 - K\mathbf{X}_2 \|_{\mathcal{S}_\mathbf{G}} &= \max_{t \in [0,\tau]} \| K\mathbf{X}_1(\cdot,t) - K\mathbf{X}_2(\cdot,t) \|_{L^\infty(I;\mathbb{R}^{1\times F})} \\
&= \max_{t \in [0,\tau]} \left\| \int_0^t \Phi(\mathbf{W}; \mathbf{X}_1(\cdot,s); \mathbf{H}(s)) - \Phi(\mathbf{W}; \mathbf{X}_2(\cdot,s); \mathbf{H}(s)) ds \right\|_{L^\infty(I;\mathbb{R}^{1\times F})} \\
&\leq \tau \max_{t \in [0,\tau]} \| \Phi(\mathbf{W}; \mathbf{X}_1(\cdot,t); \mathbf{H}(t)) - \Phi(\mathbf{W}; \mathbf{X}_2(\cdot,t); \mathbf{H}(t)) \|_{L^\infty(I;\mathbb{R}^{1\times F})}. \quad (26)
\end{aligned}
$$

It follows from Lemma A.1 that

$$\|\Phi(\mathbf{W}; \mathbf{X}_1(\cdot, t); \mathbf{H}(t)) - \Phi(\mathbf{W}; \mathbf{X}_2(\cdot, t); \mathbf{H}(t))\|_{L^\infty(I; \mathbb{R}^{1 \times F})} \leq (FM_1)^L \|\mathbf{X}_1(\cdot, t) - \mathbf{X}_2(\cdot, t)\|_{L^\infty(I; \mathbb{R}^{1 \times F})}$$

where the constant $M_1$ is defined in 23. By substituting the above estimate into 26, we obtain that

$$\|K\mathbf{X}_1 - K\mathbf{X}_2\|_{\mathcal{S}_\mathbf{G}} \leq \tau(FM_1)^L \max_{t \in [0,\tau]} \|\mathbf{X}_1(\cdot, t) - \mathbf{X}_2(\cdot, t)\|_{L^\infty(I; \mathbb{R}^{1 \times F})}$$

$$= \tau(FM_1)^L \|\mathbf{X}_1 - \mathbf{X}_2\|_{\mathcal{S}_\mathbf{G}} \leq \frac{1}{2} \|\mathbf{X}_1 - \mathbf{X}_2\|_{\mathcal{S}_\mathbf{G}}$$

where the last inequality follows from the definition of $\tau$. Therefore, the operator $K$ is a contraction. By the Banach contraction mapping principle, there exists an unique solution $\bar{\mathbf{X}} \in \mathcal{S}_\mathbf{G}$ of the IVP 6. Taking $\bar{\mathbf{X}}(\tau)$ as the initial condition, we repeat the argument to extend the solution to $[0, 2\tau]$. In such a way, we can keep doing until the solution extends to $[0, T]$, and get a unique solution $\mathbf{X} \in C([0, T]; L^\infty(I; \mathbb{R}^{1 \times F}))$. According to AS0 and AS1, it follows that $\Phi(\mathbf{W}; \mathbf{X}(u, \cdot); \mathbf{H}(\cdot))$ is continuous, that is, the integrand in 25 is continuous. Therefore, by fundamental theorem of calculus, we see that $K\mathbf{X}$ is continuously differentiable about the second variable $t$. As $K\mathbf{X} = \mathbf{X}$, we conclude that $\mathbf{X} \in C^1([0, T]; L^\infty(I; \mathbb{R}^{1 \times F}))$. This completes the proof. $\qquad\square$

## A.5 TRANSFERABILITY OF SOLUTIONS FOR GRAPHON-NDES

In this subsection, we prove that the solution of 15 converges to the solution of 6 as the number of nodes tends to infinity. Such result serves as the basis for proving our main result Theorems 3.2 and 3.3. We proceed with several helpful lemmas for showing the transferability of graphon neural networks.

**Lemma A.2** (Werner (2006)Satz VII.1.4). *Let $\mathcal{H}$ be a Hilbert space and $A \in \mathcal{B}(\mathcal{H})$ be normal. Let $h : \mathbb{R} \to \mathbb{C}$ be a continuous function, then $\|h(A)\|_{\mathcal{H} \to \mathcal{H}} = \|\chi_{\sigma(A)} h\|_{L^\infty(\mathbb{R})}$, where $\chi_{\sigma(A)}$ is the indicator function of $\sigma(A)$.*

Suppose that $\mathcal{H}$ is a separable Hilbert space and an operator $A : \mathcal{H} \to \mathcal{H}$ is compact and self-adjoint. For $1 \leq p \leq \infty$, the Schatten $p$-norm $\| \cdot \|_{\mathcal{S}_p}$ of the operator $A$ is defined as the $\ell_p$ norm of the sequence of its eigenvalues. It is known (Theorem 7.3 Weidmann (2012)) that the $\ell_\infty$ norm of the sequence of its eigenvalues equals to the operator norm of $A$. Therefore, it holds that $\|A\|_{\mathcal{S}_\infty} = \|A\|_{\mathrm{op}}$. The following statement is a direct corollary of the monotonicity of $\ell_p$ norm.

**Lemma A.3.** *Suppose that $\mathcal{H}$ is a separable Hilbert space and an operator $A : \mathcal{H} \to \mathcal{H}$ is compact and self-adjoint. Then, for $1 \leq p \leq p' \leq \infty$, $\|A\|_{\mathcal{S}_1} \geq \|A\|_{\mathcal{S}_p} \geq \|A\|_{\mathcal{S}_{p'}} \geq \|A\|_{\mathrm{op}} = \|A\|_{\mathcal{S}_\infty}$.*

**Lemma A.4** (Potapov & Sukochev (2011)Theorem 1,Maskey et al. (2023)Lemma A6). *Let $\mathcal{H}$ be a Hilbert space and $\mathcal{S}(\mathcal{H}) \subset \mathcal{B}(\mathcal{H})$ be the space of bounded self-adjoint operators with the operator norm topology. Let $h : \mathbb{R} \to \mathbb{C}$ be a continuous function. Then the mapping*

$$\mathcal{S}(\mathcal{H}) \to \mathcal{B}(\mathcal{H}), A \to h(A)$$

*is continuous. In addition, if function $h$ is Lipchitz continuous on $\mathbb{R}$ with Lipchistz constant $\mathrm{Lip}(h)$, then for $p \in (1, \infty)$, there exits a constant $K_p > 0$ such that $\|h(A) - h(B)\|_{\mathcal{S}_p} \leq \mathrm{Lip}(h) K_p \|A - B\|_{\mathcal{S}_p}$ for any self-adjoint operators $A$ and $B$ satisfying $\|A - B\|_{\mathcal{S}_p} < \infty$.*

**Lemma A.5.** *Let $T > 0$ and graphon $\mathbf{W} \in L^\infty(I^2)$. If AS0 holds, then the function $J_\mathbf{W} : [0, T] \to \mathbb{R}$ defined by*

$$J_\mathbf{W}(t) := \max_{\ell \in [L]} \sqrt{\sum_{f, g \in [F]} \left\| \mathbf{h}_{\ell, t}^{fg}(\cdot) \chi_{\sigma(T_\mathbf{W})}(\cdot) \right\|_{L^\infty(\mathbb{R})}^2}, \quad t \in [0, T], \qquad (27)$$

*is Lipschitz continuous with $\mathrm{Lip}(J_\mathbf{W}(\cdot)) = FKA_0$.*

*Proof.* Since $\sigma(T_\mathbf{W}) \subset [-\|T_\mathbf{W}\|_{\mathrm{op}}, \|T_\mathbf{W}\|_{\mathrm{op}}]$ and $\|T_\mathbf{W}\|_{\mathrm{op}} \leq \|T_\mathbf{W}\|_{\mathcal{S}_2} = \|T_\mathbf{W}\|_{\mathrm{HS}} = \|\mathbf{W}\|_{L^2(I^2)} \leq 1$, we have $\sigma(T_\mathbf{W}) \subset [-1, 1]$ and hence function $J_\mathbf{W}$ is well-defined. Let $t_1, t_2 \in [0, T]$ be arbitrary but

fixed. It follows from the reverse triangle inequalities of norms $\|\cdot\|_\infty$, $\|\cdot\|_2$ and $L^\infty(\mathbb{R})$ that

$$
|J_{\mathbf{W}}(t_2) - J_{\mathbf{W}}(t_1)| \leq \max_{\ell \in [L]} \left| \sqrt{\sum_{f,g \in [F]} \left\| \mathbf{h}_{\ell,t_2}^{fg}(\cdot)\chi_{\sigma(T_{\mathbf{W}})}(\cdot) \right\|_{L^\infty(\mathbb{R})}^2} - \sqrt{\sum_{f,g \in [F]} \left\| \mathbf{h}_{\ell,t_1}^{fg}(\cdot)\chi_{\sigma(T_{\mathbf{W}})}(\cdot) \right\|_{L^\infty(\mathbb{R})}^2} \right|
$$

$$
\leq \max_{\ell \in [L]} \sqrt{\sum_{f,g \in [F]} \left( \left\| \mathbf{h}_{\ell,t_2}^{fg}(\cdot)\chi_{\sigma(T_{\mathbf{W}})}(\cdot) \right\|_{L^\infty(\mathbb{R})} - \left\| \mathbf{h}_{\ell,t_1}^{fg}(\cdot)\chi_{\sigma(T_{\mathbf{W}})}(\cdot) \right\|_{L^\infty(\mathbb{R})} \right)^2}
$$

$$
\leq \max_{\ell \in [L]} \sqrt{\sum_{f,g \in [F]} \left\| \left( \mathbf{h}_{\ell,t_2}^{fg}(\cdot) - \mathbf{h}_{\ell,t_1}^{fg}(\cdot) \right) \chi_{\sigma(T_{\mathbf{W}})}(\cdot) \right\|_{L^\infty(\mathbb{R})}^2}
$$

$$
= \max_{\ell \in [L]} \sqrt{\sum_{f,g \in [F]} \left\| \left( \sum_{k=0}^{K-1} \left( [\boldsymbol{h}_\ell^{fg}]_k(t_2) - [\boldsymbol{h}_\ell^{fg}]_k(t_1) \right)(\cdot)^k \right) \chi_{\sigma(T_{\mathbf{W}})}(\cdot) \right\|_{L^\infty(\mathbb{R})}^2}
$$

$$
\leq \max_{\ell \in [L]} \sqrt{\sum_{f,g \in [F]} \left( \sum_{k=0}^{K-1} \left| [\boldsymbol{h}_\ell^{fg}]_k(t_2) - [\boldsymbol{h}_\ell^{fg}]_k(t_1) \right| \left\| (\cdot)^k \chi_{\sigma(T_{\mathbf{W}})}(\cdot) \right\|_{L^\infty(\mathbb{R})} \right)^2}
$$

where the equation follows from 7 and the last inequality follows from the triangle inequality of norm $\|\cdot\|_{L^\infty(\mathbb{R})}$. According to $\sigma(T_{\mathbf{W}}) \subset [-1,1]$, we have $\left\| (\cdot)^k \chi_{\sigma(T_{\mathbf{W}})}(\cdot) \right\|_{L^\infty(\mathbb{R})} \leq 1$ for all $k \in [K]$. Therefore, with assumption AS0, we have

$$
|J_{\mathbf{W}}(t_2) - J_{\mathbf{W}}(t_1)| \leq \max_{\ell \in [L]} \sqrt{\sum_{f,g \in [F]} \left( \sum_{k=0}^{K-1} \left| [\boldsymbol{h}_\ell^{fg}]_k(t_2) - [\boldsymbol{h}_\ell^{fg}]_k(t_1) \right| \right)^2}
$$

$$
\leq \max_{\ell \in [L]} \sqrt{\sum_{f,g \in [F]} \left( \sum_{k=0}^{K-1} A_0 |t_2 - t_1| \right)^2} = FKA_0|t_2 - t_1|.
$$

Therefore, $J_{\mathbf{W}}$ is Lipschitz continuous with $\mathrm{Lip}(J_{\mathbf{W}}) = FKA_0$. $\qquad\square$

We remark that the Lipschitz constant in Lemma A.5 is independent of the graphon $\mathbf{W}$. This implies that $J_{\mathbf{W}}$ is equicontinuous for different graphons $\mathbf{W}$.

**Lemma A.6.** *Let $T > 0$, $\mathbf{X} \in C([0,T]; L^\infty(I; \mathbb{R}^{1 \times F}))$, and graphon $\mathbf{W} \in L^\infty(I^2)$. Let function $J_{\mathbf{W}}$ be defined by 27. If AS0 and AS1 hold, then*

$$
\|\Phi(\mathbf{W}; \mathbf{X}(\cdot,t); \mathbf{H}(t))\|_{L^2(I; \mathbb{R}^{1 \times F})} \leq M_{\mathbf{W}}^L \|\mathbf{X}(\cdot,t)\|_{L^2(I; \mathbb{R}^{1 \times F})},
$$

*where*

$$
M_{\mathbf{W}} := \sup_{t \in [0,T]} J_{\mathbf{W}}(t). \tag{28}
$$

*Proof.* With assumption AS0, according to Lemma A.5, the function $J_{\mathbf{W}}$ is continuous, and hence the supremum $M_{\mathbf{W}}$ can be obtained over the interval $[0,T]$. Let $t \in [0,T]$ be arbitrary but fixed. Let $\mathbf{X}_{0,t} := \mathbf{X}(\cdot,t)$. For each $f \in [F]$, we denote by $\mathbf{X}_{\ell,t} = [X_{\ell,t}^f : f \in [F]]$ the output function of $\ell$-th layer in Graphon neural network with input feature functions $\mathbf{X}_{0,t}$. It is clear that for each $\ell \in [L]$, we have $\mathbf{X}_{\ell,t} \in L^\infty(I; \mathbb{R}^{1 \times F})$. It follow from AS1 that the activation function $\rho$ satisfies

$$
|\rho(x)| \leq |x|, \quad x \in \mathbb{R}. \tag{29}
$$

For $f \in [F]$, $\ell \in [L]$, according to the updating formula 5 and inequality 29, we have

$$
\left\| X_{\ell,t}^f \right\|_{L^2(I)} = \left\| \rho \left( \sum_{g=1}^F \mathbf{h}_{\ell,t}^{fg}(T_{\mathbf{W}}) X_{\ell-1,t}^g \right) \right\|_{L^2(I)} \leq \left\| \sum_{g=1}^F \mathbf{h}_{\ell,t}^{fg}(T_{\mathbf{W}}) X_{\ell-1,t}^g \right\|_{L^2(I)}
$$

$$
\leq \sum_{g=1}^F \left\| \mathbf{h}_{\ell,t}^{fg}(T_{\mathbf{W}}) \right\|_{L^2(I) \to L^2(I)} \left\| X_{\ell-1,t}^g \right\|_{L^2(I)} \leq \sqrt{\sum_{g=1}^F \left\| \mathbf{h}_{\ell,t}^{fg}(T_{\mathbf{W}}) \right\|_{L^2(I) \to L^2(I)}^2} \left\| \mathbf{X}_{\ell-1,t} \right\|_{L^2(I;\mathbb{R}^{1 \times F})},
$$

Therefore,

$$
\|\mathbf{X}_{\ell,t}\|_{L^2(I;\mathbb{R}^{1 \times F})}^2 \leq \sum_{f=1}^F \sum_{g=1}^F \left\| \mathbf{h}_{\ell,t}^{fg}(T_{\mathbf{W}}) \right\|_{L^2(I) \to L^2(I)}^2 \|\mathbf{X}_{\ell-1,t}\|_{L^2(I;\mathbb{R}^{1 \times F})}^2 \tag{30}
$$

It follows from Lemma A.2 that $\left\| \mathbf{h}_{\ell,t}^{fg}(T_{\mathbf{W}}) \right\|_{L^2(I) \to L^2(I)} = \left\| \mathbf{h}_{\ell,t}^{fg}(\cdot) \chi_{\sigma(T_{\mathbf{W}})}(\cdot) \right\|_{L^\infty(I)}$. Then, with $M_{\mathbf{W}}$ defined by 28, we have

$$
\sum_{f=1}^F \sum_{g=1}^F \left\| \mathbf{h}_{\ell,t}^{fg}(T_{\mathbf{W}}) \right\|_{L^2(I) \to L^2(I)}^2 \leq M_{\mathbf{W}}^2. \tag{31}
$$

Substituting 31 into 30, we obtain that

$$
\|\mathbf{X}_{\ell,t}\|_{L^2(I;\mathbb{R}^{1 \times F})}^2 \leq M_{\mathbf{W}}^2 \|\mathbf{X}_{\ell-1,t}\|_{L^2(I;\mathbb{R}^{1 \times F})}^2,
$$

followed by

$$
\|\mathbf{X}_{L,t}\|_{L^2(I;\mathbb{R}^{1 \times F})} \leq M_{\mathbf{W}}^L \|\mathbf{X}_{0,t}\|_{L^2(I;\mathbb{R}^{1 \times F})}.
$$

The proof is complete by noting that $\mathbf{X}_{L,t} = \Phi(\mathbf{W}; \mathbf{X}(\cdot, t); \mathbf{H}(t))$ and $\mathbf{X}_{0,t} = \mathbf{X}(\cdot, t)$. $\qquad\square$

Below, we show transferability of graphon neural networks.

**Lemma A.7.** *Let $T > 0$, $\mathbf{X}_1, \mathbf{X}_2 \in C([0,T]; L^\infty(I; \mathbb{R}^{1 \times F}))$, and graphons $\mathbf{W}_1, \mathbf{W}_2 \in L^\infty(I^2)$. If AS1 and AS2 hold, then for any $1 < p < \infty$, we have*

$$
\|\Phi(\mathbf{W}_1; \mathbf{X}_1(\cdot, t); \mathbf{H}(t)) - \Phi(\mathbf{W}_2; \mathbf{X}_2(\cdot, t); \mathbf{H}(t))\|_{L^2(I;\mathbb{R}^{1 \times F})}
$$

$$
\leq (\sqrt{2}M)^L \|\mathbf{X}_1(\cdot, t) - \mathbf{X}_2(\cdot, t)\|_{L^2(I;\mathbb{R}^{1 \times F})} + \sqrt{2}^{L+1} L_{\mathbf{H},p} M^{L-1} \|T_{\mathbf{W}_1} - T_{\mathbf{W}_2}\|_{\mathcal{S}_p} \|\mathbf{X}_2(\cdot, t)\|_{L^2(I;\mathbb{R}^{1 \times F})}
$$

*where*

$$
C_{\mathbf{H}} := \max_{t \in [0,T], \ell \in [L]} \sqrt{\sum_{f,g \in [F]} \left\| \mathbf{h}_{\ell,t}^{fg}(\cdot) \chi_{[-1,1]}(\cdot) \right\|_{L^\infty(\mathbb{R})}^2}, \tag{32}
$$

*and*

$$
L_{\mathbf{H},p} := \max_{t \in [0,T], \ell \in [L]} K_p \sqrt{\sum_{f,g \in [F]} \mathrm{Lip}^2(\mathbf{h}_{\ell,t}^{fg}(\cdot))}. \tag{33}
$$

*Proof.* Let $t \in [0, T]$ be arbitrary but fixed. Let $\mathbf{X}_{0,t} := \mathbf{X}_1(\cdot, t)$ and $\mathbf{Y}_{0,t} := \mathbf{X}_2(\cdot, t)$. For each $f \in [F]$, we denote by $\mathbf{X}_{\ell,t} = [X_{\ell,t}^f : f \in [F]]$, $\mathbf{Y}_{\ell,t} = [Y_{\ell,t}^f : f \in [F]]$ the output function of $\ell$-th layer in Graphon neural network with input feature functions $\mathbf{X}_{0,t}$ and $\mathbf{Y}_{0,t}$, respectively. It is clear that for each $\ell \in [L]$, we have $\mathbf{X}_{\ell,t}, \mathbf{Y}_{\ell,t} \in L^\infty(I; \mathbb{R}^{1 \times F})$. For $f \in [F]$, $\ell \in [L]$, according to the

updating formula 5 and AS1, we have

$$
\begin{aligned}
\left\| X_{\ell,t}^f - Y_{\ell,t}^f \right\|_{L^2(I)} &= \left\| \rho\left( \sum_{g=1}^F \mathbf{h}_{\ell,t}^{fg}(T_{\mathbf{W}_1}) X_{\ell-1,t}^g \right) - \rho\left( \sum_{g=1}^F \mathbf{h}_{\ell,t}^{fg}(T_{\mathbf{W}_2}) Y_{\ell-1,t}^g \right) \right\|_{L^2(I)} \\
&\leq \left\| \sum_{g=1}^F \mathbf{h}_{\ell,t}^{fg}(T_{\mathbf{W}_1}) X_{\ell-1,t}^g - \sum_{g=1}^F \mathbf{h}_{\ell,t}^{fg}(T_{\mathbf{W}_2}) Y_{\ell-1,t}^g \right\|_{L^2(I)} \\
&\leq \sum_{g=1}^F \left\| \mathbf{h}_{\ell,t}^{fg}(T_{\mathbf{W}_1}) \right\|_{L^2(I)\to L^2(I)} \left\| X_{\ell-1,t}^g - Y_{\ell-1,t}^g \right\|_{L^2(I)} \\
&\quad + \sum_{g=1}^F \left\| \mathbf{h}_{\ell,t}^{fg}(T_{\mathbf{W}_1}) - \mathbf{h}_{\ell,t}^{fg}(T_{\mathbf{W}_2}) \right\|_{L^2(I)\to L^2(I)} \left\| Y_{\ell-1,t}^g \right\|_{L^2(I)} \\
&\leq \sqrt{ \sum_{g=1}^F \left\| \mathbf{h}_{\ell,t}^{fg}(T_{\mathbf{W}_1}) \right\|_{L^2(I)\to L^2(I)}^2 } \, \|\mathbf{X}_{\ell-1,t} - \mathbf{Y}_{\ell-1,t}\|_{L^2(I;\mathbb{R}^{1\times F})} \\
&\quad + \sqrt{ \sum_{g=1}^F \left\| \mathbf{h}_{\ell,t}^{fg}(T_{\mathbf{W}_1}) - \mathbf{h}_{\ell,t}^{fg}(T_{\mathbf{W}_2}) \right\|_{L^2(I)\to L^2(I)}^2 } \, \|\mathbf{Y}_{\ell-1,t}\|_{L^2(I;\mathbb{R}^{1\times F})}
\end{aligned}
$$

Therefore,

$$
\|\mathbf{X}_{\ell,t} - \mathbf{Y}_{\ell,t}\|_{L^2(I;\mathbb{R}^{1\times F})}^2 \leq 2\sum_{f=1}^F\sum_{g=1}^F \left\| \mathbf{h}_{\ell,t}^{fg}(T_{\mathbf{W}_1}) \right\|_{L^2(I)\to L^2(I)}^2 \|\mathbf{X}_{\ell-1,t} - \mathbf{Y}_{\ell-1,t}\|_{L^2(I;\mathbb{R}^{1\times F})}^2 \tag{34}
$$

$$
+ 2\sum_{f=1}^F\sum_{g=1}^F \left\| \mathbf{h}_{\ell,t}^{fg}(T_{\mathbf{W}_1}) - \mathbf{h}_{\ell,t}^{fg}(T_{\mathbf{W}_2}) \right\|_{L^2(I)\to L^2(I)}^2 \|\mathbf{Y}_{\ell-1,t}\|_{L^2(I;\mathbb{R}^{1\times F})}^2 \tag{35}
$$

It follows from Lemma A.2 that $\left\| \mathbf{h}_{\ell,t}^{fg}(T_{\mathbf{W}_1}) \right\|_{L^2(I)\to L^2(I)} = \left\| \mathbf{h}_{\ell,t}^{fg}(\cdot)\chi_{\sigma(T_{\mathbf{W}_1})}(\cdot) \right\|_{L^\infty(I)}$. Note that $\sigma(T_{\mathbf{W}_1}) \subset [-1,1]$. Then, with $C_{\mathbf{H}}$ defined by 32, we have

$$
\sum_{f=1}^F\sum_{g=1}^F \left\| \mathbf{h}_{\ell,t}^{fg}(T_{\mathbf{W}_1}) \right\|_{L^2(I)\to L^2(I)}^2 \leq C_{\mathbf{H}}^2 \tag{36}
$$

According to Lemma A.6 and again $\sigma(T_{\mathbf{W}_2}) \subset [-1,1]$, we obtain that

$$
\|\mathbf{Y}_\ell\|_{L^2(I;\mathbb{R}^{1\times F})}^2 \leq C_{\mathbf{H}}^{2\ell} \|\mathbf{Y}_{0,t}\|_{L^2(I;\mathbb{R}^{1\times F})}^2. \tag{37}
$$

Additionally, it follows from AS2, Lemmas A.4 and A.3 that there exists $K_p > 0$ such that

$$
\left\| \mathbf{h}_{\ell,t}^{fg}(T_{\mathbf{W}_1}) - \mathbf{h}_{\ell,t}^{fg}(T_{\mathbf{W}_2}) \right\|_{L^2(I)\to L^2(I)} \leq \left\| \mathbf{h}_{\ell,t}^{fg}(T_{\mathbf{W}_1}) - \mathbf{h}_{\ell,t}^{fg}(T_{\mathbf{W}_2}) \right\|_{\mathcal{S}_p} \leq K_p \mathrm{Lip}\left( \mathbf{h}_{\ell,t}^{fg}(\cdot) \right) \|T_{\mathbf{W}_1} - T_{\mathbf{W}_2}\|_{\mathcal{S}_p}.
$$

Then, with $L_{\mathbf{H},p}$ defined by 33, we have

$$
\sum_{f=1}^F\sum_{g=1}^F \left\| \mathbf{h}_{\ell,t}^{fg}(T_{\mathbf{W}_1}) - \mathbf{h}_{\ell,t}^{fg}(T_{\mathbf{W}_2}) \right\|_{L^2(I)\to L^2(I)}^2 \leq L_{\mathbf{H},p}^2 \|T_{\mathbf{W}_1} - T_{\mathbf{W}_2}\|_{\mathcal{S}_p}^2. \tag{38}
$$

Plugging the estimates 36, 37, 38 into 35, with $M$ defined by 32, we obtain that

$$
\|\mathbf{X}_{\ell,t} - \mathbf{Y}_{\ell,t}\|_{L^2(I;\mathbb{R}^{1\times F})}^2 \leq 2C_{\mathbf{H}}^2 \|\mathbf{X}_{\ell-1,t} - \mathbf{Y}_{\ell-1,t}\|_{L^2(I;\mathbb{R}^{1\times F})}^2 + 2L_{\mathbf{H},p}^2 \|T_{\mathbf{W}_1} - T_{\mathbf{W}_2}\|_{\mathcal{S}_p}^2 C_{\mathbf{H}}^{2\ell-2} \|\mathbf{Y}_{0,t}\|_{L^2(I;\mathbb{R}^{1\times F})}^2.
$$

Solving the recurrence equation, we obtain that

$$
\|\mathbf{X}_{L,t} - \mathbf{Y}_{L,t}\|_{L^2(I;\mathbb{R}^{1\times F})}^2 \leq (2C_{\mathbf{H}}^2)^L \|\mathbf{X}_{0,t} - \mathbf{Y}_{0,t}\|_{L^2(I;\mathbb{R}^{1\times F})}^2 + 2L_{\mathbf{H},p}^2 \|T_{\mathbf{W}_1} - T_{\mathbf{W}_2}\|_{\mathcal{S}_p}^2 \|\mathbf{Y}_{0,t}\|_{L^2(I;\mathbb{R}^{1\times F})}^2 C_{\mathbf{H}}^{2(L-1)}(2^L - 1).
$$

yielding

$$\|\mathbf{X}_{L,t}-\mathbf{Y}_{L,t}\|_{L^2(I;\mathbb{R}^{1\times F})} \leq (\sqrt{2}C_{\mathbf{H}})^L\|\mathbf{X}_{0,t}-\mathbf{Y}_{0,t}\|_{L^2(I;\mathbb{R}^{1\times F})}+\sqrt{2}^{L+1}L_{\mathbf{H},p}\|T_{\mathbf{W}_1}-T_{\mathbf{W}_2}\|_{\mathcal{S}_p}\|\mathbf{Y}_{0,t}\|_{L^2(I;\mathbb{R}^{1\times F})}C_{\mathbf{H}}^{L-1}.$$

By noting that $\mathbf{X}_{L,t} = \Phi(\mathbf{W}_1;\mathbf{X}_1(\cdot,t);\mathbf{H}(t))$, $\mathbf{Y}_{L,t} = \Phi(\mathbf{W}_2;\mathbf{X}_2(\cdot,t);\mathbf{H}(t))$, $\mathbf{X}_{0,t} = \mathbf{X}_1(\cdot,t)$ and $\mathbf{Y}_{0,t} = \mathbf{X}_2(\cdot,t)$, we obtain the desired result from the above inequality. $\square$

The following lemma is a direct consequence of Arzelà–Ascoli theorem.

**Lemma A.8.** *Let $\Omega \subset \mathbb{R}$ be an interval. Let $\{f_n : \Omega \to \mathbb{R}\}_{n\in\mathbb{N}}$ be an equicontinuous sequence of functions. If $\lim_{n\to\infty} f_n(x) = f(x)$, for all $x \in \Omega$, then the sequence $\{f_n\}_{n\in\mathbb{N}}$ uniformly converges to $f$.*

**Lemma A.9.** *Suppose that AS0-AS2 hold. Let $T > 0$, graphon $\mathbf{W}_n$ and initial graphon feature function $\mathbf{G}_n$ be defined by 13 and 14, respectively. Let $J_{\mathbf{W}_n}$ be defined by 27 with $\mathbf{W}$ replaced by $\mathbf{W}_n$. Let $\mathbf{X}_n$ be the solution of 15. Then it holds that*

$$\|\mathbf{X}_n\|_{C([0,T];L^2(I;\mathbb{R}^{1\times F}))} \leq \|\mathbf{G}_n\|_{L^2(I;\mathbb{R}^{1\times F})} \exp(M_n^L T), \tag{39}$$

*where*

$$M_n := \sup_{t\in[0,T]} J_{\mathbf{W}_n}(t). \tag{40}$$

*Moreover, if*

$$\lim_{n\to\infty} \|\mathbf{G}_n - \mathbf{G}\|_{L^2(I;\mathbb{R}^{1\times F})} = 0, \quad \lim_{n\to\infty} \|T_{\mathbf{W}_n} - T_{\mathbf{W}}\|_{L^2(I)\to L^2(I)} = 0, \tag{41}$$

*then there exists a positive constant $B$ (depending on $\mathbf{W}$, $\mathbf{G}$, $F$, $K$, $L$ and $T$) such that*

$$\|\mathbf{X}_n\|_{C([0,T];L^2(I;\mathbb{R}^{1\times F}))} \leq B. \tag{42}$$

*Proof.* Let $t \in [0,T]$ be arbitrary but fixed. It follows from 47 that

$$\frac{1}{2}\frac{\partial}{\partial t}\|\mathbf{X}_n(\cdot,t)\|_{L^2(I;\mathbb{R}^{1\times F})}^2 = \left|\int_I \frac{\partial\mathbf{X}_n(u,t)^\top}{\partial t}\mathbf{X}_n(u,t)du\right| = \left|\int_I \Phi(\mathbf{W}_n;\mathbf{X}_n(u,t);\mathbf{H}(t))^\top\mathbf{X}_n(u,t)du\right|$$

$$\leq \|\Phi(\mathbf{W}_n;\mathbf{X}_n(\cdot,t);\mathbf{H}(t))\|_{L^2(I;\mathbb{R}^{1\times F})}\|\mathbf{X}_n(\cdot,t)\|_{L^2(I;\mathbb{R}^{1\times F})}$$

which combining with Lemma A.6 and constant $M_n$ defined by 40 yields

$$\frac{\partial}{\partial t}\|\mathbf{X}_n(\cdot,t)\|_{L^2(I;\mathbb{R}^{1\times F})}^2 \leq 2M_n^L\|\mathbf{X}_n(\cdot,t)\|_{L^2(I;\mathbb{R}^{1\times F})}^2.$$

It follows from Grönwall's inequality that for all $t \in [0,T]$,

$$\|\mathbf{X}_n(\cdot,t)\|_{L^2(I;\mathbb{R}^{1\times F})}^2 \leq \|\mathbf{X}_n(\cdot,0)\|_{L^2(I;\mathbb{R}^{1\times F})}^2 \exp(2M_n^L T),$$

which combining with the initial condition of 15 yields

$$\|\mathbf{X}_n\|_{C([0,T];L^2(I;\mathbb{R}^{1\times F}))} = \sup_{t\in[0,T]} \|\mathbf{X}_n(\cdot,t)\|_{L^2(I;\mathbb{R}^{1\times F})}^2 \leq \|\mathbf{G}_n\|_{L^2(I;\mathbb{R}^{1\times F})}^2 \exp(2M_n^L T),$$

which proves 39. Moreover, if 41 holds, then

$$\lim_{n\to\infty} \|\mathbf{G}_n\|_{L^2(I;\mathbb{R}^{1\times F})} = \|\mathbf{G}\|_{L^2(I;\mathbb{R}^{1\times F})} \tag{43}$$

and for any $f, g \in [F]$, $\ell \in [L]$ and $t \in [0,T]$, it follows from Lemma A.4 and continuity of the function $\mathbf{h}_{\ell,t}^{fg}$ that

$$\lim_{n\to\infty} \left\|\mathbf{h}_{\ell,t}^{fg}(T_{\mathbf{W}_n}) - \mathbf{h}_{\ell,t}^{fg}(T_{\mathbf{W}})\right\|_{L^2(I)\to L^2(I)} = 0,$$

and hence

$$\lim_{n\to\infty} \left\|\mathbf{h}_{\ell,t}^{fg}(T_{\mathbf{W}_n})\right\|_{L^2(I)\to L^2(I)} = \left\|\mathbf{h}_{\ell,t}^{fg}(T_{\mathbf{W}})\right\|_{L^2(I)\to L^2(I)}.$$

According to Lemma A.2, we obtain that

$$\lim_{n\to\infty} \left\|\mathbf{h}_{\ell,t}^{fg}(\cdot)\chi_{\sigma(T_{\mathbf{W}_n})}(\cdot)\right\|_{L^\infty(\mathbb{R})} = \left\|\mathbf{h}_{\ell,t}^{fg}(\cdot)\chi_{\sigma(T_{\mathbf{W}})}(\cdot)\right\|_{L^\infty(\mathbb{R})}.$$

This implies that for each $t \in [0,T]$, $\lim_{n\to\infty} J_{\mathbf{W}_n}(t) = J_{\mathbf{W}}(t)$. Moreover, we notice from Lemma A.5 that $J_{\mathbf{W}}$ and $J_{\mathbf{W}_n}$, $n \in \mathbb{N}$ are Lipschitz continuous functions with the same Lipschitz constant $FKA_0$. And hence the sequence $\{J_{\mathbf{W}_n}\}_{n\in\mathbb{N}}$ is equicontinuous. Therefore, according to Lemma A.8, we obtain that $\{J_{\mathbf{W}_n}\}_{n\in\mathbb{N}}$ uniformly converges to $J_{\mathbf{W}}$. That is, for any $\epsilon > 0$, there exists $N > 0$, such that when $n > N$, for all $t \in [0,T]$, it holds that $|J_{\mathbf{W}_n}(t) - J_{\mathbf{W}}(t)| < \epsilon$, or

$$\sup_{t\in[0,T]} |J_{\mathbf{W}_n}(t) - J_{\mathbf{W}}(t)| < \epsilon.$$

Therefore, we have

$$|M_{\mathbf{W}_n} - M_{\mathbf{W}}| = \left| \sup_{t\in[0,T]} J_{\mathbf{W}_n}(t) - \sup_{t\in[0,T]} J_{\mathbf{W}}(t) \right| \leq \sup_{t\in[0,T]} |J_{\mathbf{W}_n}(t) - J_{\mathbf{W}}(t)| < \epsilon.$$

This implies that

$$\lim_{n\to\infty} M_{\mathbf{W}_n} = M_{\mathbf{W}}. \tag{44}$$

Combining 43 and 44, we conclude that there exists a positive constant $B$ (depending on $\mathbf{W}$, $\mathbf{G}$, $F$, $K$, $L$ and $T$) such that for all $n \in \mathbb{N}$, there holds

$$\|\mathbf{G}_n\|_{L^2(I;\mathbb{R}^{1\times F})} \exp(M_n^L T) \leq B,$$

which combining with 39 yields 42. $\qquad\square$

We remark that condition 41 is satisfied for Models I and II. In fact, we will further present the specific convergence rates of 41 in next subsection.

**Lemma A.10** (Generalized Gronwall's inequality). *Let $a$, $b$ and $c$ be non-negative constants. Let $u(t)$ be a non-negative function that satisfies the integral inequality*

$$u(t) \leq c + \int_0^t \left( au(s) + bu^{\frac{1}{2}}(s) \right) ds,$$

*then we have*

$$u(t) \leq \left( c^{\frac{1}{2}} \exp(at/2) + \frac{\exp(at/2) - 1}{a} b \right)^2.$$

*Proof.* This result is a special case of Perov (1959) (also see Theorem 21 in Dragomir (2003)). $\qquad\square$

We now show that the solution $\mathbf{X}_n$ of IVP 15 converges to the solution $\mathbf{X}$ of IVP 6 as the number $n$ of nodes goes to infinity.

**Theorem A.11.** *Suppose that AS0-AS2 hold. Let $\mathbf{X}$ and $\mathbf{X}_n$ denote the solutions of 6 and 15, respectively. Suppose that 41 holds, and let $B$ be the constant appearing in 42. Then for any $p > 1$, it holds that*

$$\|\mathbf{X} - \mathbf{X}_n\|_{C([0,T];L^2(I;\mathbb{R}^{1\times F}))} \leq P\|\mathbf{G} - \mathbf{G}_n\|_{L^2(I;\mathbb{R}^{1\times F})} + Q\|T_{\mathbf{W}} - T_{\mathbf{W}_n}\|_{\mathcal{S}_p}, \tag{45}$$

*where*

$$P := \exp((\sqrt{2}C_{\mathbf{H}})^L T), \quad Q := \frac{\exp((\sqrt{2}C_{\mathbf{H}})^L T) - 1}{C_{\mathbf{H}}^L} \sqrt{2} L_{\mathbf{H},p} C_{\mathbf{H}}^{L-1} B. \tag{46}$$

*Proof.* Let $t \in [0,T]$ be arbitrary but fixed. Denote $\mathbf{Z}_n = \mathbf{X} - \mathbf{X}_n$. By subtracting 15 from 6, we have

$$\frac{\partial}{\partial t} \mathbf{Z}_n(u,t) = \Phi(\mathbf{W}; \mathbf{X}(u,t); \mathbf{H}(t)) - \Phi(\mathbf{W}_n; \mathbf{X}_n(u,t); \mathbf{H}(t)). \tag{47}$$

It follows from 47 that

$$\frac{1}{2}\frac{\partial}{\partial t} \|\mathbf{Z}_n(\cdot,t)\|^2_{L^2(I;\mathbb{R}^{1\times F})} = \left| \int_I \frac{\partial \mathbf{Z}_n(u,t)^\top}{\partial t} \mathbf{Z}_n(u,t) du \right|$$

$$= \left| \int_I (\Phi(\mathbf{W}; \mathbf{X}(u,t); \mathbf{H}(t)) - \Phi(\mathbf{W}_n; \mathbf{X}_n(u,t); \mathbf{H}(t)))^\top \mathbf{Z}_n(u,t) du \right|$$

$$\leq \|\Phi(\mathbf{W}; \mathbf{X}(\cdot,t); \mathbf{H}(t)) - \Phi(\mathbf{W}_n; \mathbf{X}_n(\cdot,t); \mathbf{H}(t))\|_{L^2(I;\mathbb{R}^{1\times F})} \|\mathbf{Z}_n(\cdot,t)\|_{L^2(I;\mathbb{R}^{1\times F})}$$

$$\tag{48}$$

It follows from Lemmas A.7 and A.9 that

$$\|\Phi(\mathbf{W}; \mathbf{X}(\cdot, t); \mathbf{H}(t)) - \Phi(\mathbf{W}_n; \mathbf{X}_n(\cdot, t); \mathbf{H}(t))\|_{L^2(I; \mathbb{R}^{1 \times F})}$$

$$\leq (\sqrt{2}C_{\mathbf{H}})^L \|\mathbf{Z}_n(\cdot, t)\|_{L^2(I; \mathbb{R}^{1 \times F})} + \sqrt{2}^{L+1} L_{\mathbf{H},p} C_{\mathbf{H}}^{L-1} \|T_{\mathbf{W}} - T_{\mathbf{W}_n}\|_{\mathcal{S}_p} B. \qquad (49)$$

Let $u(t) := \|\mathbf{Z}(\cdot, t)\|_{L^2(I; \mathbb{R}^{1 \times F})}^2$, $a := 2(\sqrt{2}C_{\mathbf{H}})^L$, $b := 2\sqrt{2}^{L+1} L_{\mathbf{H},p} C_{\mathbf{H}}^{L-1} \|T_{\mathbf{W}} - T_{\mathbf{W}_n}\|_{\mathcal{S}_p} B$. With the introduced notation, we combine the estimates 48 and 49, and obtain that

$$\frac{\partial}{\partial t} u(t) \leq a u(t) + b u^{\frac{1}{2}}(t).$$

Let $T' \in [0, T]$ be arbtrary but fixed. We integrate above $[0, T']$ about the variable $t$, and get

$$u(T') \leq u(0) + \int_0^{T'} \left( a u(t) + b u^{\frac{1}{2}}(t) \right) dt.$$

We then apply Lemma A.10, and hence

$$u(T') \leq \left( u(0)^{\frac{1}{2}} \exp(aT'/2) + \frac{\exp(aT'/2) - 1}{a} b \right)^2 \leq \left( u(0)^{\frac{1}{2}} \exp(aT/2) + \frac{\exp(aT/2) - 1}{a} b \right)^2.$$

With constants $P$ and $Q$ defined in 46, and recalling the definition of $u(t)$, we have

$$\|\mathbf{Z}_n(\cdot, T')\|_{L^2(I; \mathbb{R}^{1 \times F})} \leq P \|\mathbf{Z}_n(\cdot, 0)\|_{L^2(I; \mathbb{R}^{1 \times F})} + Q \|T_{\mathbf{W}} - T_{\mathbf{W}_n}\|_{\mathcal{S}_p}.$$

Noting that $\mathbf{Z}_n(\cdot, t) = \mathbf{X}(\cdot, t) - \mathbf{X}_n(\cdot, t)$, $\mathbf{Z}_n(\cdot, 0) = \mathbf{G} - \mathbf{G}_n$ and the arbitrariness of $T' \in [0, T]$, we further obtain that

$$\sup_{t \in [0,T]} \|\mathbf{X}(\cdot, t) - \mathbf{X}_n(\cdot, t)\|_{L^2(I; \mathbb{R}^{1 \times F})} \leq P \|\mathbf{G} - \mathbf{G}_n\|_{L^2(I; \mathbb{R}^{1 \times F})} + Q \|T_{\mathbf{W}} - T_{\mathbf{W}_n}\|_{\mathcal{S}_p}.$$

The proof is complete by noting that

$$\|\mathbf{X} - \mathbf{X}_n\|_{C([0,T]; L^2(I; \mathbb{R}^{1 \times F}))} = \sup_{t \in [0,T]} \|\mathbf{X}(\cdot, t) - \mathbf{X}_n(\cdot, t)\|_{L^2(I; \mathbb{R}^{1 \times F})}.$$

$\square$

## A.6 Convergence rate

In this section, we establish the convergence rate of the solution $X_n$ of 15 converging to the solution $X$ of 6. This is done by employing inequality 45 with $p = 2$. It follows from Conway (1990) that Schatten 2-norm is Hilbert-Schmidt norm, moreover for integral operator, there holds

$$\|T_{\mathbf{W}_n} - T_{\mathbf{W}_n}\|_{\mathcal{S}_2} = \|\mathbf{W} - \mathbf{W}_n\|_{L^2(I^2)}.$$

Therefore, inequality 45 with $p = 2$ gives

$$\|\mathbf{X} - \mathbf{X}_n\|_{C([0,T]; L^2(I; \mathbb{R}^{1 \times F}))} \leq P \|\mathbf{G} - \mathbf{G}_n\|_{L^2(I; \mathbb{R}^{1 \times F})} + Q \|\mathbf{W} - \mathbf{W}_n\|_{L^2(I^2)}. \qquad (50)$$

The above estimate indicates that it suffices to analyze the convergence rates of $\|\mathbf{G} - \mathbf{G}_n\|_{L^2(I; \mathbb{R}^{1 \times F})}$ and $\|\mathbf{W} - \mathbf{W}_n\|_{L^2(I^2)}$ in order to determine that of $\|\mathbf{X} - \mathbf{X}_n\|_{C([0,T]; L^2(I; \mathbb{R}^{1 \times F}))}$.

*proof of Theorem 3.2.* We set $u_i := (i - 1)/n$, $I_i := [u_i, u_{i+1}]$, for each $i \in [n]$. According to definition $\mathbf{W}_n$ of 13 with 8, we have

$$\|\mathbf{W} - \mathbf{W}_n\|_{L^2(I^2)}^2 = \int_{I^2} |\mathbf{W}(u, v) - \mathbf{W}_n(u, v)|^2 \, dudv$$

$$= \sum_{i,j \in [n]} \int_{I_i \times I_j} |\mathbf{W}(u, v) - \mathbf{W}_n(u, v)|^2 \, dudv$$

$$= \sum_{i,j \in [n]} \int_{I_i \times I_j} |\mathbf{W}(u, v) - \mathbf{W}(u_i, u_j)|^2 \, dudv$$

According to AS3, we obtain that

$$\|\mathbf{W} - \mathbf{W}_n\|_{L^2(I^2)}^2 \le A_1^2 \sum_{i,j \in [n]} \int_{I_i \times I_j} (|x - u_i| + |y - u_j|)^2 \, du dv \tag{51}$$

For each $i, j \in [n]$, direct computation gives

$$\int_{I_i \times I_j} (|u - u_i| + |v - u_j|)^2 \, du dv = \left(\frac{2}{3} + \frac{1}{4}\right) \frac{1}{n^4} = \frac{11}{12n^4}$$

which combining with 51 gives

$$\|\mathbf{W} - \mathbf{W}_n\|_{L^2(I^2)}^2 \le n^2 \frac{11 A_1^2}{12 n^4} = \frac{11 A_1^2}{12 n^2}. \tag{52}$$

We recall that $\mathbf{G} = [G_f : f \in [F]]$ and $\mathbf{G}_n = [(G_n)_f : f \in [F]]$, and hence

$$\|\mathbf{G} - \mathbf{G}_n\|_{L^2(I;\mathbb{R}^{1 \times F})}^2 = \sum_{f \in [F]} \|G_f - (G_n)_f\|_{L^2(I)}^2. \tag{53}$$

For each $f \in [F]$, according to definition $\mathbf{G}_n$ of 14 with 9, we have

$$\|G_f - (G_n)_f\|_{L^2(I)}^2 = \int_I |G_f(u) - (G_n)_f(u)|^2 du = \sum_{j \in [n]} \int_{I_j} |G_f(u) - (G_n)_f(u)|^2 du$$

$$= \sum_{j \in [n]} \int_{I_j} |G_f(u) - G_f(u_j)|^2 du. \tag{54}$$

It follows from AS4 that for each $j \in [n]$, there holds

$$\int_{I_j} |G_f(u) - G_f(u_j)|^2 du \le A_2^2 \int_{I_j} (u - u_j)^2 du = \frac{A_2^2}{3 n^3}.$$

We substitute the above estimate into 54, and obtain that

$$\|G_f - (G_n)_f\|_{L^2(I)}^2 \le n \frac{A_2^2}{3 n^3} = \frac{A_2^2}{3 n^2}.$$

Therefore, from 53, we have

$$\|\mathbf{G} - \mathbf{G}_n\|_{L^2(I;\mathbb{R}^{1 \times F})}^2 \le \frac{A_2^2 F}{3 n^2}. \tag{55}$$

We substitute estimates 52 and 55 into 50, and get

$$\|\mathbf{X} - \mathbf{X}_n\|_{C([0,T];L^2(I;\mathbb{R}^{1 \times F}))} \le P \|\mathbf{G} - \mathbf{G}_n\|_{L^2(I;\mathbb{R}^{1 \times F})} + Q \|\mathbf{W} - \mathbf{W}_n\|_{L^2(I^2)}$$

$$\le \left(\sqrt{\frac{F}{3}} A_2 P + \sqrt{\frac{11}{12}} A_1 Q\right) \frac{1}{n}$$

By letting

$$C := \sqrt{\frac{F}{3}} A_2 P + \sqrt{\frac{11}{12}} A_1 Q$$

$$= \sqrt{\frac{F}{3}} A_2 \exp((\sqrt{2} C_{\mathsf{H}})^L T) + \sqrt{\frac{11}{12}} A_1 \frac{\exp((\sqrt{2} C_{\mathsf{H}})^L T) - 1}{C_{\mathsf{H}}^L} \sqrt{2} L_{\mathsf{H},2} C_{\mathsf{H}}^{L-1} B \tag{56}$$

where $C_{\mathsf{H}}$, $L_{\mathsf{H},2}$ are defined by 32 and 33 with $p = 2$, respectively, the proof is complete.

$\square$

We proceed to examine the convergence rate of graphons for simple graphs. It is important to note that in this case, assumption AS3 is not met. For a simple graphon $\mathbf{W}$, we recall that the support of $\mathbf{W}$ is denoted by $\mathbf{W}^+$. To estimate $\|\mathbf{W} - \mathbf{W}_n\|_{L^2(I^2)}$ appearing in 50, it is necessary to describe the complexity of the boundary $\partial\mathbf{W}^+$ of the support $\mathbf{W}^+$. We achieve this by utilizing the upper box-counting dimension from fractal geometry Falconer (2014). Let $F$ be any non-empty bounded subset of $\mathbb{R}^2$ and let $\mathcal{N}_\delta(F)$ be the number of $\delta$-mesh cubes that intersect $F$. The upper box-counting dimensions of $F$ is defined as

$$\overline{\dim}_\mathrm{B} F := \varlimsup_{\delta \to 0} \frac{\log \mathcal{N}_\delta(F)}{-\log \delta}. \tag{57}$$

It is clear that $\overline{\dim}_\mathrm{B} F \in [0, 2]$ for any non-empty bounded subset $F$ of $\mathbb{R}^2$. As a simple example, the straight line $\{(x, 0) : x \in [0, 1]\}$ has an upper box-counting dimension of 1. More intricate curves will have a larger upper box-counting dimension. Therefore, it is reasonable to assume that $\overline{\dim}_\mathrm{B}(\partial\mathbf{W}^+) \in [1, 2)$.

**Lemma A.12.** *Suppose that $\Omega \subset \mathbb{R}^d$ and $f \in L^2(\Omega)$. Let $|\Omega|$ be the volume of $\Omega$. Then the constant function*

$$h(u) := \frac{1}{|\Omega|} \int_\Omega f(u) du, \quad u \in \Omega,$$

*is the best constant approximation of $f$, that is,*

$$\inf\{\|f - c\|_{L^2(\Omega)} : c \in \mathbb{R}\} = \|f - h\|_{L^2(\Omega)}. \tag{58}$$

*Proof.* For any $c \in \mathbb{R}$, we have

$$\|f - c\|_{L^2(\Omega)}^2 = \int_\Omega |f(u) - c|^2 du = |\Omega|c^2 - 2\left(\int_\Omega f(u) du\right) c + \int_\Omega |f(u)|^2 du \tag{59}$$

which is a quadratic function about $c$. Therefore, the minimum is obtained at

$$c = \frac{1}{|\Omega|} \int_\Omega f(u) du.$$

The proof is complete by noting that the minimizers of $\|f - c\|_{L^2(\Omega)}^2$ and $\|f - c\|_{L^2(\Omega)}$ are the same. $\qquad\square$

*proof of Theorem 3.3.* We begin with estimating $\|\mathbf{W} - \mathbf{W}_n\|_{L^2(I^2)}$. Recall that $\mathcal{N}_\delta(\partial\mathbf{W}^+)$ denotes the number of $\delta$-mesh cubes that intersect $\partial\mathbf{W}^+$. We set $\delta = 1/n$, and it follows from definition of $\mathbf{W}_n$ that

$$\|\mathbf{W} - \mathbf{W}_n\|_{L^2(I^2)}^2 = \int_I |\mathbf{W}(u, v) - \mathbf{W}_n(u, v)|^2 dudv \le \mathcal{N}_{1/n}(\partial\mathbf{W}^+) \frac{1}{n^2}. \tag{60}$$

According to definition 57 of upper box-counting dimension, for any $\epsilon > 0$, there exists $N_\mathbf{W} \in \mathbb{N}$ such that when $n > N_\mathbf{W}$, $\frac{\log \mathcal{N}_{1/n}(\partial\mathbf{W}^+)}{-\log(1/n)} < b + \epsilon$. Therefore, $\mathcal{N}_{1/n}(\partial\mathbf{W}^+) \le n^{b+\epsilon}$ which combining with 60 yields

$$\|\mathbf{W} - \mathbf{W}_n\|_{L^2(I^2)} \le n^{-(1 - \frac{b+\epsilon}{2})}. \tag{61}$$

We proceed to estimate $\|\mathbf{G} - \mathbf{G}_n\|_{L^2(I;\mathbb{R}^{1\times F})}$. Let $\mathbf{G}'_n$ defined by 14 in the way of 9. It has been shown in the proof of Theorem 3.2 that, with assumption AS4,

$$\|\mathbf{G} - \mathbf{G}'_n\|_{L^2(I;\mathbb{R}^{1\times F})} \le A_2 \sqrt{\frac{F}{3}} \frac{1}{n}. \tag{62}$$

According to Lemma A.12, we obtain that

$$\|\mathbf{G} - \mathbf{G}_n\|_{L^2(I;\mathbb{R}^{1\times F})}^2 = \sum_{f \in [F]} \|G_f - (G_n)_f\|_{L^2(I)}^2 \le \sum_{f \in [F]} \|G_f - (G'_n)_f\|_{L^2(I)}^2 = \|\mathbf{G} - \mathbf{G}'_n\|_{L^2(I;\mathbb{R}^{1\times F})}^2$$

which combining with 62 implies

$$\|\mathbf{G} - \mathbf{G}_n\|_{L^2(I;\mathbb{R}^{1\times F})} \le A_2 \sqrt{\frac{F}{3}} \frac{1}{n}. \tag{63}$$

We substitute estimates 61 and 63 into 50, and get

$$\|\mathbf{X} - \mathbf{X}_n\|_{C([0,T];L^2(I;\mathbb{R}^{1\times F}))} \le P\|\mathbf{G} - \mathbf{G}_n\|_{L^2(I;\mathbb{R}^{1\times F})} + Q\|\mathbf{W} - \mathbf{W}_n\|_{L^2(I^2)}$$
$$\le \left(PA_2\sqrt{\frac{F}{3}} + Q\right)n^{-(1-\frac{b+\epsilon}{2})}$$

By letting

$$\widetilde{C} := PA_2\sqrt{\frac{F}{3}} + Q$$
$$= \exp((\sqrt{2}C_{\mathbf{H}})^L T)A_2\sqrt{\frac{F}{3}} + \frac{\exp((\sqrt{2}C_{\mathbf{H}})^L T) - 1}{C_{\mathbf{H}}^L}\sqrt{2}L_{\mathbf{H},2}C_{\mathbf{H}}^{L-1}B \tag{64}$$

where $C_{\mathbf{H}}$, $L_{\mathbf{H},2}$ are defined by 32 and 33 with $p = 2$, respectively, the proof is complete.

$\square$