# OpenReview forum: "Graphon Neural Differential Equations and Transferabilty of Graph Neural Differential Equations"
_ICLR.cc/2025/Conference — Submitted to ICLR 2025_

### Official Review · Reviewer_Du2B · 2024-11-03

**Soundness:** 2
**Presentation:** 3
**Contribution:** 2
**Rating:** 5
**Confidence:** 2

**Summary:**

This paper considers the transferability of Graph Neural ODEs (G-NODEs) in the sense of how much difference two solutions of G-NODEs derived from different graphs have. First, the Graphon-ODE, a time-continuous version of the Graphon GNN, is formulated, and its well-posedness is shown.
Next, two variants of Graphon-ODE (Models I and II) are considered for a graph sequence made by spatially discretizing a graphon $W$. The approximation error between each solution of the Graphon-ODEs and the solution of Graphon-ODE derived from the original graphon is evaluated. As a corollary, the difference between the solutions of graphon-ODEs derived from two sequence graphs converging to the same graphon was evaluated.
The types of numerical experiments verify the validity of the theoretical analyses:
1. The approximation ability of models learned on small graphs by simulating the heat equation on large graphs.
2. The dependence of the relative error on the box dimension on the CheckerBoard Graphon.
3. Performance of node prediction tasks of models trained with subgraphs on whole graphs using the Cora citation network.

**Strengths:**

- As far as I know, this is the first study to extend Graph Neural ODEs to graphons.
- The background knowledge on graphons is adequately described, making the paper accessible for readers unfamiliar with the graph limit theory.

**Weaknesses:**

- I have questions about whether the setup and evaluation methods of the numerical experiments are appropriate to claim that support the correctness of the theoretical analyses.
- I have questions about the interpretation of the experimental results (Table 1).
- The clarity of the paper may be improved. Several similar concepts appear, making it difficult to understand the relationship between them. Also, the description of experimental settings has room for improvement.

See Questions for details.

**Questions:**

- Theorem 3.2 shows that the solution of a Graphon ODE induced from $\mathcal{G}\_{n}$ that converges to graphon $W$ approximates the Graphon ODE induced from $W$. We cannot take arbitrary sequence $\mathcal{G}\_{n}$ that converges to $W$, but specifically constructs a specific convergence sequence $\mathcal{G}\_{n}$ from graphon $W$ (Models I, II). Also, we need to know $W$ to construct $\mathcal{G}\_{n}$. These assumptions limit the applicability of the theory.
- This paper compares the solution $X_n(t)$ of the GNDE (12) with the solution $X(\cdot, t)$ of the Graphon-CNDE (6) by transforming (12) into the Graphon-CNDE (15), solve (15), and compare its solution $X_n(\cdot, t)$ with that of (6). While this is one way to compare the solutions, since both are Graphon-NODE solutions, I think it is not appropriate to say that it quantifies the approximation error between GNDE and Graphon-ODE, as described in the abstract. Wouldn't it be more natural to regard the solution $X_n(T)$ of (12) as a graphon by interpreting it as a piecewise-constant function?
- What is the definition of the true dynamics $Y_N(1)$ in the numerical experiments? Is it the solution of the heat equation in (18) made by a numerical simulation method? In that case, I have a question about whether it is appropriate to call it *true dynamics* since the simulation is an approximate numerical solution. Also, where does the dependence on $N$ come from?
- In Figure 3, the authors claim that *GNDEs can learn complex physical dynamics on smaller graphs [...] and effectively transfer this knowledge to larger systems [...].* from the fact the relative error is around $10^{-2}$. I have a question about whether this interpretation is appropriate. First, the theoretical bound $O(1/500)$ is an abuse of the O-nation. Second, it needs to be clarified why we can claim the relative error $10^{-2}$ is sufficiently small without comparison with other approximated solutions.
- The meaning of *transferability* seems to differ between theoretical analyses and numerical experiments. If I do not miss any information, the theoretical analyses do not explicitly define transferability. As far as we can infer from Theorem 3.3, it refers to the difference between the outputs of Graphon-NODEs on small and large graphs is small. On the other hand, in the experiments in Section *Transfer Learning of Nonlinear Heat Equations on Complete Weighted Graphs*, transferability means that true dynamics and the model outputs are close. Therefore, whether the numerical experiment results validate the theoretical analyses is questionable.
- How are *the GNDE outputs on the original and subgraphs* calculated in numerical experiments on $\{0, 1\}$-valued Checkerboad graphon? In particular, what do original and subgraph mean, respectively?
- Which test data are used for *Subgraph Accuracy* and *Full Graph Accuracy* in Table 1 respectively? If I do not miss any information, *Subgraph Accuracy* is not mentioned in the text. For *Full Graph Accuracy*, does it correspond to the *test accuracy for the full dataset* in the text?
- The paper draws the following conclusion from Table 1: *As we train on a larger proportion of nodes, we gain accuracy on full graph prediction*. However, I do not think it is appropriate for the following reasons. First, since the standard deviation of the results in Table 1 is large, it is difficult to say that there is a significant difference between Subgraph Accuracy and Full Graph Accuracy. Second, even if we ignore the standard deviation, the Subgraph accuracy value exceeds the full graph accuracy at 20%. As the percentage increases over 20%, the difference between Subgraph accuracy and Full graph accuracy decreases.
- Can we interpret the subsampling method in the Cora experiments as Model I or II in the theoretical analysis? If so, an explanation should be provided for the justification.


**Minor Comments**

- Citations are not appropriate; citep and citet should be used appropriately. For example, *this framework generalizes Neural ODEs Chen et al. (2018)* in Section 1 should be *this framework generalizes Neural ODEs (Chen et al., 2018)*.
- The paper writes that *One of the most prevalent GNN architectures is the GCNs, introduced by Bruna et al. (2013) [...]*. However, the reference Bruna et al. (2013) does not use the term *graph convolution* nor GCN. Also, GCN is considered a proper term for the architecture introduced by Kipf and Welling (2016). Therefore, I think it is more appropriate to write like *GCNs, whose origin can be traced back to Bruna et al. (2013) and popularized by Kipf & Welling (2016).*
- The homomorphism count $\mathrm{hom}(\mathcal{F}, \mathcal{G})$ is undefined.
- The homomorphism density $t(\mathcal{F}, \mathbf{W})$ is undefined for the pair of a graph $\mathcal{F}$ and a graphon $\mathbf{W}$.
- Many similar concepts related to neural ODEs appear, such as GNDE (1), GCNDE (4), Graphon-CNDE (6), and Graphon-NDE induced by GNDE (15). Since it is difficult to understand their relationships at first reading, I would suggest clarifying them by, e.g., making diagrams to show their relationships.
- $G(u)$ is in $L^{\infty}(I; \mathbb{R}^{1\times F})$ in Eq. (6). However, the graphon convolution assumes $L^2$-integrability in the previous section. It should be commented that $G(u)$ is $L^2$ (we can verify it because $L^\infty \subset L^2$ using the fact that $I$ is compact.) Also, rigorously speaking about well-definedness, it should be shown that $X(\cdot, t)$ is in $L^{\infty}$ for any $t$.
- In Theorem 3.1, it should be made clear what IVP stands for (Initial Value Problem?)
- It is not easy to read when formulas are referenced without prefixes (e.g., IVP 6, solution of 15). It is preferable to use a form such as Eq. (6) to clarify that it references a formula.
- P8: *We mention that [...], so in general, AS1 is not satifsfied for Model II.*: AS1 -> AS3?
- It is better to add a reference to Adam.
- *The relative error [...] is shown in Figure 3 (a).*: Figure 3 does not have sub-items such as (a).
- Dormand-Price -> Dormand-Prince

**Details Of Ethics Concerns:**

N.A.

---

> ### Author Response · Authors · 2024-11-26
> **Response to Reviewer Du2B**
>
> Thanks for carefully reading our paper! We will incorporate your suggestions in the future revision. Below we show several responses to key points in the paper.
>
> ### Questions
>
> - Theorem 3.2 shows that the solution of a Graphon ODE induced from $\mathcal{G}_n$ that converges to graphon $W$ approximates the Graphon ODE induced from $W$. We cannot take an arbitrary sequence $\mathcal{G}_n$ that converges to $W$, but instead, we must specifically construct a sequence $\mathcal{G}_n$ from graphon $W$ (Models I, II). Additionally, we need to know $W$ to construct $\mathcal{G}_n$. These assumptions limit the applicability of the theory.
>
> **Response**: In the revision, we will present a new theorem using the current techniques to demonstrate the convergence of any convergent graph sequence, not limited to those sampled from a graphon. This will broaden the scope and applicability of our theory. Thank you for pointing out it!
>
> - This paper compares the solution $X_n(t)$ of the GNDE (12) with the solution $X(\dot,t)$ of the Graphon-CNDE (6) by transforming (12) into the Graphon-CNDE (15), solve (15), and compare its solution $X_n(\dot,t)$ with that of (6). While this is one way to compare the solutions, since both are Graphon-NODE solutions, I think it is not appropriate to say that it quantifies the approximation error between GNDE and Graphon-ODE, as described in the abstract. Wouldn't it be more natural to regard the solution $X_n(T)$ of (12) as a graphon by interpreting it as a piecewise-constant function?
>
> **Response**: In fact, the two ways you described are the same. We will include a lemma to show it in the future revision.
>
>
> - Experiments
>
> **Response**: we will make clear descriptions and improve it in the future.

---

> > ### Comment · Reviewer_Du2B · 2024-12-02
> >
> > I thank the authors for the response and am sorry for the late response.
> >
> > **Questions**
> >
> > > In the revision, we will present a new theorem using the current techniques to demonstrate the convergence of any convergent graph sequence, not limited to those sampled from a graphon.
> >
> > > In fact, the two ways you described are the same. We will include a lemma to show it in the future revision.
> >
> > OK. When I receive the new results related to these two responses, I will check them.
> >
> >
> > **Experiments**
> >
> > > we will make clear descriptions and improve it in the future.
> >
> > OK

---

### Official Review · Reviewer_HvLg · 2024-11-03

**Soundness:** 3
**Presentation:** 1
**Contribution:** 1
**Rating:** 3
**Confidence:** 4

**Summary:**

This paper proposes a study on the transferability of neuro graph ODEs through the perspective of graphons. The key idea is to present conditions for when can transferability happen, based on the concept of graph limits.
The authors define graphon neuro ODEs, and then present several experiments to verify their method.

**Strengths:**

1.  **Relevance:** The question of the paper is interesting. It is important to know and quantify the transferability of GNNs.

2. **Correness:** The method itself seems to be correct.

**Weaknesses:**

1. **Related works**: A significant lack of discussion of relevant works is missing in this paper. From neuro ODEs [1-2], to graph neuro ODEs [3-8]. Also, in terms of continuous ODE based GNNs, the authors ignore works like [9-11]. This raises doubts about how thoroughly the study proposed in this paper was done.

2. **Claims with no proofs/references:** In the introduction section (section 1), the authors make major claims such as "Recent advances have introduced Graphon Neural Networks (Graphon-NNs) as limit objects of GNNs, establishing theoretical bounds on the approximation error between GNNs and their corresponding Graphon-NNs. These results reveal a fundamental trade-off between discriminability and transferability." However the authors do not provide references/proofs of these claims.

3. **Limited use of GNNs**: The authors focus on the GCN architecture, which is limiting and does not show whether the proposed method can work with other graph neural architectures.

4. **Definition of Graphon**: Since this paper revolves around graphons, I find that the late formal introduction of graphons in the paper makes it hard to follow.

5. **Low quality of presentation:** The paper suffers from an overall lack of quality in terms of its presentation. For example, references are broken in the sense that they do not tell to which element references are (e.g., Equation, Section, etc.). Also, the paper is hard to follow and should be significantly edited to be pleasant to read and easy to follow.

6. **Missing comparison with Graph Transformer:** The authors consider the case of a fully connected graph with edge weights. This is almost identical to Graph Transformers, see [12-15] for examples. I would expect the authors to discuss and compare these methods.

7. **Missing discussion on computational cost, and potentitally high complexity:** The complexity of the method is not discussed in the paper. Morever, to the best of my understanding the method is also built on the use of a fully-connected graph, which makes it very expensive. Can the authors please elaborate?

8.  **Limited and unconvincing experiments:** The experiments provided in this paper are simple and not convincing. That is due to several reasons:
A. The experiment of heat diffusion is rather simplistic, and does not really show that any transferable knowledge was studied. Employing the diffusion equation on different graph sizes will yield the same process, so the result shown here is not surprising.
B. The results on Cora look very low compared to standard results on this dataset (which are usually around 80% accuracy). I understand that the authors use a subgraph of the Cora network to train the GNN, but it is well-known that even simple diffusion can yield strong results on this dataset. Therefore I am not convinced that the results are valid, and code is not provided, so it is hard to understand how the results were obtained.
C. While the experiments in A and B are simple, they are welcome given that they work. However, to show the transferability of learned models, more experiments are required. For example, recent papers on GNNs show multiple benchmarks on graph transferability in [16-18].


**References:**

[1] Stable Architectures for Deep Neural Networks

[2] A Proposal on Machine Learning via Dynamical Systems

[3] GRAND: Graph Neural Diffusion

[4] PDE-GCN: Novel Architectures for Graph Neural Networks Motivated by Partial Differential Equations

[5] GRAND++: Graph Neural Diffusion with A Source Term

[6] GREAD: Graph Neural Reaction-Diffusion Networks

[7] Graph-Coupled Oscillator Networks

[8] Unleashing the Potential of Fractional Calculus in Graph Neural Networks with FROND

[9] Monotone Operator Theory-Inspired Message Passing for Learning Long-Range Interaction on Graphs

[10] Implicit graph neural networks: a monotone operator viewpoint

[11] Long Range Propagation on Continuous-Time Dynamic Graphs

[12] Graph Transformer Networks

[13] Attending to Graph Transformers

[14] Recipe for a General, Powerful, Scalable Graph Transformer

[15] A Generalization of Transformer Networks to Graphs

[16] AnyGraph: Graph Foundation Model in the Wild

[17] GraphAny: A Foundation Model for Node Classification on Any Graph

[18] Transfer learning with graph neural networks for improved molecular property prediction in the multi-fidelity setting

**Questions:**

Please see in the enlisted weaknesses.

---

> ### Author Response · Authors · 2024-11-26
> **Response to Reviewer  HvLg**
>
> **Clarification on Transferability and Transfer Learning**
>
> - We would like to clarify that the concept of transferability in our paper is distinct from the transfer learning that the reviewer may have in mind. Specifically, we focus on the post-training regime, where the hyperparameters are fixed, and the primary variable is the graph structure. Our study explores how to evaluate and compare the outputs of two different Graph Neural Differential Equations (GNDEs) operating on distinct graph structures. This focus allows us to analyze and establish theoretical foundations in this context.
>
>
> **Response to Weakness 6**
>
> - We feel that a comparison with graph transformers is unnecessary and outside the scope of our work. As stated in the manuscript, we are not proposing a new architecture but are conducting a theoretical investigation into convolutional structures. To our knowledge, this is the first paper that extends Graph Neural ODEs to graphons, as pointed out by Reviewer Du2B. The focus on graphon settings provides unique insights that are independent of transformer-based methods, which serve a different purpose.
>
> **Response to Weakness 7**
>
> - Our work assumes that hyperparameters are fixed and given, emphasizing the evaluation phase rather than hyperparameter tuning. This setup aligns with relevant transferability study in  GNN literature.
>
>
> **Response to Weakness 8**
>
> - We appreciate the reviewer’s feedback and would like to seek further input regarding our experimental validation. In our study, we conducted the following experiments:
> 1. Example 1 demonstrates that the bound $1/n$ is sharp, providing theoretical validation for our complexity analysis.
> 2. Example 2 illustrates the effectiveness of our proposed complexity measure in characterizing the convergence speed.
> 3. Example 3 shows the transferability of the hyperparameter obtained from subgraphs to the performance on the full graph on real world data-sets.
>
> After fixing a bug in the Cora code, we now achieve full graph accuracy matching results reported in other papers. Furthermore, we observed convergence results on additional real world datasets using GCNDEs.
>
> Given the transferability problem we studied , we would like to know whether this set of experiments sufficiently addresses the reviewer’s concerns, or if additional experiments are necessary to strengthen the paper.

---

> > ### Comment · Reviewer_HvLg · 2024-11-29
> >
> > Thank you authors for the response.
> >
> > Regarding weakness 6, I think that graph transformers are relevant here. Just because the paper does not offer a new architecture (which is fine) does not mean you should not compare with relevant methods.
> >
> >
> > Regarding weakness 7, your response does not answer my point. I was asking about the computational complexity of the method.
> >
> > Regarding weakness 8, I have provided you with references and suggestions in my review; please read it and try to use it in subsequent versions of your paper.
> >
> > Regarding 'finding bugs in code', I am sorry, but this is not confidence-inspiring, and it is not the way we should do experiments. I would expect full transparency and a revision to the paper, which, as claimed by the authors, is not planned for this round of reviewers. In particular, you need to explain what the bug is, what the results are, and what the re-evaluation of other methods looks like after this modification.
> >
> > Lastly, the authors did not address all the comments given in my review. Importantly, please do not ignore previous works in the fields you are working on (please see in my review), and provide evidence to your claims.
> >
> >
> > I hope that the authors will implement some of the suggestions and comments in future versions of their paper.

---

> > > ### Author Response · Authors · 2024-11-29
> > >
> > > Thank you for your feedback. We will make every effort to incorporate relevant references and make our experiments transparent in future revisions. However, we would like to clarify some fundamental misunderstandings regarding our work:
> > >
> > > 1. Purpose of Our Method: Our analytic method is designed to compare solutions of GNDEs on two different graphs using graphons as a theoretical tool. Graph transformers are architectural frameworks, we can not understand the meaning of comparing graph transformers with our method, as graph transformer is not a way of comparing solutions of GNDEs on two different graphs.
> > >
> > > 2. Computational Complexity: Our method is not algorithmic and does not have computational complexity in the traditional sense: it is a purely analytical approach based on dynamical system and graphon theory, with fixed hyperparameters. Do you mean reporting computation complexity of training a GNDEs?

---

> > > > ### Comment · Reviewer_HvLg · 2024-11-30
> > > >
> > > > Dear authors,
> > > >
> > > > Thank you for the response. It is true that your method inspects the solution of GNDEs. My intention in point 1 was that you should also check its behavior when using GNDEs that use transformers, like in [1]. Regarding point 2, although the method proposed here is not about training a new network, I assume that computational costs are incurred. Please correct me if I am wrong. Thus, my comment was that I would like to know what the complexity of the explainability required here. Let me give you an example: suppose you were to perform eigen-decomposition of weight matrices. Clearly, this has a large cost and may not be scalable, depending on the architecture we consider. Thus I think it would be useful if the authors discuss this point in their paper.
> > > >
> > > >
> > > >
> > > >
> > > > [1] Advective Diffusion Transformers for Topological Generalization in Graph Learning

---

> > > > > ### Author Response · Authors · 2024-11-30
> > > > > **Response to Reviewer HvLg**
> > > > >
> > > > > Dear Reviewer  HvLg,
> > > > >
> > > > > Thanks for your response.
> > > > >
> > > > > 1. **Scope of Our Paper**:
> > > > > Our paper establishes a convergence theory for graph convolutional neural differential equations (GCNDEs). Extending such analysis to other architectures, like graph transformers, would require dedicated numerical convergence studies as a preliminary step for theoretical development. However, we believe such an exploration is beyond the scope of this work and should be done in future papers addressing such problems.
> > > > >
> > > > > **Difference with [1]**:  The focus of [1] is to advocate Advective Diffusion Transformers  in improving robustness to topological shifts, which necessitates comparative studies with other architectures. In contrast, our work focuses on establishing the convergence properties of graph convolutional neural differential equations and its continuous limit, a well-established architecture. We do not claim superiority of them over other architectures and do not see the relevance of comparisons in our context.
> > > > >
> > > > > 2. We are still unclear about the phrase "computational costs of complexity of the explainability." Could you kindly elaborate or clarify this point?

---

> > > > > > ### Comment · Reviewer_HvLg · 2024-12-01
> > > > > >
> > > > > > Dear authors,
> > > > > >
> > > > > > 1. The paper in [1] is also a GNDE. Thus I think experimenting with such architectures can make your approach more broad and general while remaining within the scope of this work.
> > > > > >
> > > > > >
> > > > > > 2. As explained in my previous response - even if your method does not propose training a new architecture, clearly, there is some computational aspect to it, correct? How do you perform your analyses? I gave an example (which may not be used in your method -- but it is in the example); if one is to inspect the learned weights of a network through some decomposition of the weight matrices, then clearly, there are computations involved. Thus, I asked that in future versions, the authors discuss the complexity of their method.

---

### Official Review · Reviewer_KiSx · 2024-11-09

**Soundness:** 3
**Presentation:** 2
**Contribution:** 2
**Rating:** 3
**Confidence:** 3

**Summary:**

This paper introduces Graphon Neural Differential Equations (Graphon-NDEs) as a continuous-depth extension of Graph Neural Differential Equations (GNDEs) to enhance transferability across graphs with shared convolutional structures. GNDEs generalize Graph Neural Networks (GNNs) to continuous-depth frameworks but face challenges when transferring learned knowledge to larger, structurally similar graphs. The authors propose Graphon-NDEs as continuum limits of GNDEs, enabling a smoother transition between discrete and continuous graph representations. Using tools from dynamical systems theory and graph limit theory, they develop a mathematical framework to quantify the approximation error between GNDEs and their Graphon-NDE counterparts, showing that this error decreases as graph size grows and providing explicit convergence rates for different graph families. Empirical validation on various graph structures, such as complete weighted graphs and checkerboard graphons, supports their theoretical results, demonstrating how structural complexity impacts transferability. This work establishes a foundational approach for scaling GNDEs to larger graphs, advancing the theoretical basis for transferability and generalization in continuous-depth graph models.

**Strengths:**

- The paper presents a well-structured and theoretically sound framework. Using rigorous mathematical tools, including dynamical systems and graph limit theories, the authors derive error bounds for GNDE and Graphon-NDE approximation, with explicit convergence rates for different graph families

- The paper is generally clear and logically structured, with a coherent flow from the introduction of GNDE limitations to the development of Graphon-NDEs and their potential to mitigate these issues.

**Weaknesses:**

- The citation format is incorrect; please ensure proper use of \citep for consistency.

- The third and fourth paragraph of the Introduction lacks relevant references to support its arguments. Adding citations here would strengthen the background context.

- Writing needs refinement for clarity. For instance, the contributions are not clearly articulated. It would be helpful to list the contributions as bullet points in the Introduction, aligning them with the theoretical sections.

- Details in notation and references to equations require more precision. For example, IVP 6 in Theorem 3.1 is unclear and could benefit from a clearer explanation or reference.

- I have concerns regarding the results in Theorem 3.3. Specifically, it appears that $b + \epsilon$ could exceed 2, and there is no explicit dependence on $\epsilon$ in the result, which needs clarification.

- The paper raises important theoretical insights but does not sufficiently address their practical relevance. For instance, while Theorem 3.3 introduces box-counting dimension to capture boundary complexity, its real-world implications are less clear. It would be beneficial if the authors included a discussion on how these theoretical bounds might affect practical performance in real applications, such as how varying graph complexity might influence training times, prediction accuracy, or the stability of the Graphon-NDEs in dynamic environments.

**Questions:**

- Could you expand the empirical validation to include more complex and widely used graph structures, such as scale-free and small-world networks?

- How does the performance of Graphon-NDEs compare with standard GNNs or GNDEs without the Graphon-NDE extension?

-  How does the structural complexity of the graph (e.g., measured by the box-counting dimension) quantitatively affect the transferability and convergence rates in practical scenarios?


- Why does Theorem 3.2 achieve a better bound with fewer assumptions compared to previous works? Could you elaborate on the specific differences in assumptions and techniques that allow for this improvement?

---

> ### Author Response · Authors · 2024-11-26
> **Response to reviewer KiSx**
>
> **Weakness**: I have concerns regarding the results in Theorem 3.3. Specifically, it appears that could exceed 2, and there is no explicit dependence on in the result, which needs clarification.
>
> - Response: We have modified the range of  $\epsilon$  to (0, 2 - b) to ensure that the exponent of  1/n  is positive. Moreover, we change the notation $N_{W}$ in Theorem 3.3 to $N_{\epsilon,W}$ to emphasize that this quantity is dependent on the choice of $\epsilon$. In fact, the  $\epsilon$ appearing in Theorem 3.3 is a pre-specified parameter that can be chosen arbitrarily small, similar to the epsilon-delta language used in defining limits. This parameter only affects the threshold $N_{\epsilon,W}$ , ensuring that the result holds when the number of nodes is sufficiently large ($n>N_{\epsilon,W}$).  Consequently, the convergence order in Theorem 3.3 can be considered as almost $1/n^{1-b/2}$. As the box counting dimension $b$ will not exceed $2$, the order $1-b/2$ of $1/n$ is always positive.
>
>
> **Weakness**: The paper raises important theoretical insights but does not sufficiently address their practical relevance.
>
> - Response:  Our study focuses on the regime where the trainable parameters are fixed, and we investigate how to compare outputs on two different graph structures. The insights gained here also motivate relevant studies in GNNs. However, we acknowledge that dynamical stability is vital to address in practical applications, we did not know if there is work addressing this problem yet.  If the reviewer is aware of related works in this area,  please let us know.
>
> **Question** : How does the performance of Graphon-NDEs compare with standard GNNs or GNDEs without the Graphon-NDE extension?
>
> - Response: Comparisons between GNNs and GNDEs have been conducted in papers where GNDEs were originally proposed. Every GNDE has a corresponding Graphon-NDE representation, where the underlying graphon is a piecewise constant function on the unit square. This representation can be viewed as a specific element within a convergent sequence, with different sequences potentially converging to distinct continuous graphon limits.
>
>   The key question is the closeness of a given GNDE to its graphon limit, which emphasizes the importance of studying the convergence rate. This connection lies at the heart of our theoretical investigation, providing insights into the behavior of GNDEs as the underlying graph size increases and linking finite models to their infinite counterparts.

---

> > ### Comment · Reviewer_KiSx · 2024-11-29
> >
> > Thanks for your response. Some of the concerns are addressed. However, I found that some of my questions or suggestions were not addressed, even though they were very easy to address. The plan to modify Theorem 3.3 looks sound. It is suggested to update the manuscript to reflect the change.

---

> > > ### Author Response · Authors · 2024-11-30
> > >
> > > Thanks for your response! We are currently re-writing our paper and will make sure addressing writing issues you pointed out and discuss practical relevance of our theorems.

---

### Author Response · Authors · 2024-11-25
**Response to all reviews**

Dear reviewers,

We sincerely thank you for taking the time to review our paper and for providing valuable feedback. We have carefully reviewed your comments and noted that certain aspects of our work may have been misunderstood, particularly by Reviewer HvLg and Reviewer KiSx.

To clarify, the primary contribution of our paper lies in **rigorously proving how solutions of graph neural ODEs converge to a solution of a PDE**, which we refer to as the graphon neural differential equation. Rather than proposing a new graph neural architecture, our work **demonstrates the convergence and stability of well-established graph neural ODEs as the number of graph nodes increases**.

Despite our best efforts and dedication, the extent of revisions required to address your feedback and improve the clarity of our manuscript makes it unlikely for us to meet the deadline for this submission cycle.

Below, we have provided responses to specific comments and would greatly appreciate any further feedback to help us improve our work.

Best,

Authors

---

> ### Comment · Reviewer_HvLg · 2024-11-26
>
> Dear authors,
>
> Thank you for the response. I am keen to read your responses to my comments. However I cannot see them. Perhaps they were not submitted?

---

### Meta-Review · Area_Chair_diuB · 2024-12-21

**Metareview:**

This paper considers Graphon Neural Differential Equations (Graphon-NDEs) to study transferability of neuro graph ODEs by taking a limit to graphon. This paper would be the first attempt to extend graph ODEs to graphon settings. The authors gave theoretical analyses such as wellposedness and approximation errors. They also conducted some numerical experiments.

The idea of extending graph ODEs to graphon is interesting. The theoretical analysis of this topic brings informative insight to the topic.
On the other hand, the paper requires substantial revision. Several mathematical notions are used without their definitions. Even if there are definitions, more careful explanations on their meaning should be given. Some mathematical reasonings are not precise. Specifically, the statement of Theorem 3.3 requires revision.
The paper is not thoroughly compared with relevant work. For example, the section of the relevant work can be more comprehensive, and novelty and significance of the theoretical results compared from exiting work can be discussed in more details.

For these reasons, this paper is not ready to be published. It requires substantial revision. Thus, I cannot recommend acceptance.

**Additional Comments On Reviewer Discussion:**

The authors addressed some concerns raised by reviewers. However, they were not convinced by the authors' rebuttal.

---

### Decision · Program_Chairs · 2025-01-22

Reject